# Overlapping functions of SIX homeoproteins during embryonic myogenesis

**Maud Wurmser**[1][☯], **Rouba Madani**[1][☯], **Nathalie Chaverot**[1][☯], **Stéphanie Backer**[1], **Matthew Borok**[2], **Matthieu Dos Santos**[1], **Glenda Comai**[3,4], **Shahragim Tajbakhsh**[3,4], **Frédéric Relaix**[2], **Marc Santolini**[5], **Ramkumar Sambasivan**[6], **Rulang Jiang**[7,8], **Pascal Maire**[1]*

**1** Université de Paris Cité, Institut Cochin, INSERM, CNRS, Paris, France, **2** Univ Paris Est Creteil, INSERM, EnvA, EFS, AP-HP, IMRB, Creteil, France, **3** Stem Cells & Development, Institut Pasteur, Paris, France, **4** CNRS UMR 3738, Institut Pasteur, Paris, France, **5** Université de Paris Cité, Interaction Data Lab, CRI Paris, INSERM. Paris, France, **6** Indian Institute of Science Education and Research (IISER) Tirupati, Tirupati, Andhra Pradesh, India, **7** Division of Developmental Biology, Cincinnati Children's Hospital Medical Center, Cincinnati, Ohio, United States of America, **8** Department of Pediatrics, University of Cincinnati College of Medicine, Cincinnati, Ohio, United States of America

☯ These authors contributed equally to this work.
* pascal.maire@inserm.fr

**Data Availability Statement:** All relevant data are within the manuscript and its Supporting Information files.

## Abstract

Four SIX homeoproteins display a combinatorial expression throughout embryonic developmental myogenesis and they modulate the expression of the myogenic regulatory factors. Here, we provide a deep characterization of their role in distinct mouse developmental territories. We showed, at the hypaxial level, that the *Six1:Six4* double knockout (dKO) somitic precursor cells adopt a smooth muscle fate and lose their myogenic identity. At the epaxial level, we demonstrated by the analysis of *Six* quadruple KO (qKO) embryos, that SIX are required for fetal myogenesis, and for the maintenance of PAX7+ progenitor cells, which differentiated prematurely and are lost by the end of fetal development in qKO embryos. Finally, we showed that *Six1* and *Six2* are required to establish craniofacial myogenesis by controlling the expression of *Myf5*. We have thus described an unknown role for SIX proteins in the control of myogenesis at different embryonic levels and refined their involvement in the genetic cascades operating at the head level and in the genesis of myogenic stem cells.

## Author summary

We demonstrate with double, triple and quadruple *Six* KO mouse embryos that specific *Six* combinations are required for proper myogenesis depending on the level of the mouse embryonic body axis. We show that the *Six1* and *Six2* genes are required for craniofacial myogenesis by controlling the engagement of unsegmented cranial paraxial mesodermal cells in the myogenic pathway. We also show that hypaxial somitic dermomyotomal cells from embryos mutant for the *Six1* and *Six4* genes are unable to engage in the skeletal muscle lineage. Last, we show that embryos mutant for the four *Six* genes expressed in the myogenic lineage exhibit a defect in self-renewal of PAX7+ stem cells present in their

**Funding:** MW is supported by the University Paris Cité (UPC), by the « Association Française contre les Myopathies » (AFM) and by the Agence Nationale pour la Recherche (ANR). RM is supported by UPC and the Fondation pour la Recherche Médicale (FRM). Financial support was provided by the AFM (AFM 17406 and AFM 16315) the Institut National de la Santé et de la Recherche Médicale (INSERM), the Centre National de la Recherche Scientifique (CNRS), and the ANR (ANR16-CE-14-0032-01 Myolinc) to PM, by the AFM 16315 to ST, and by AFM 19507 and 22946 to FR. The funders had no role in study design, data collection and analysis, decision to publish, or preparation of the manuscript.

**Competing interests:** The authors have declared that no competing interests exist.

residual muscle masses, and that SIX proteins interact directly with several enhancer elements at the *Pax7* locus to control its expression. We have thus characterized new functions of SIX proteins in the control of myogenesis at different embryonic levels and refined their involvement in the genetic cascades that govern the genesis of myogenic stem cells.

## Introduction

Embryonic myogenesis is the process by which the different muscles of the body are formed. It takes place from embryonic day 8 (E8) until birth (E18.5) in the mouse embryo [1]. This process occurs on three distinct levels: craniofacial, hypaxial (including the ventral thoracic and abdominal, as well as limb muscles), and trunk epaxial level, and is regulated by genetic cascades of events converging on the activation of the myogenic regulatory factors (MRF) known as *Myf5*, *MyoD* and *Mrf4* [2]. Their expression leads to myogenic fate acquisition of bipotent or multipotent embryonic progenitor cells at those distinct levels [3]. *Myf5* is the earliest MRF expressed, and its expression is regulated by distinct enhancer elements that are controlled by upstream regulators specific for each embryonic territory [4–6].

At the axial and limb bud levels, skeletal muscles have a somitic origin and the onset of MRF expression is under the control of *Pax3*, *Pax7*, *Six1* and *Six4* [3,7]. The early expression of *Myf5* in the epaxial cells of the somite is controlled by the synergy between GLI/Hedgehog [8, 9], LEF/Wnt [10,11] and RBPJ/Notch [12] signaling pathways, while its early expression in the hypaxial dermomyotome is controlled by the synergy between atypical Wnt signaling [13], PAX3 [14] and SIX1/4 homeoproteins [6,15,16]. The formation of the limb bud muscles requires the migration of the PAX3+SIX1+ progenitor cells from the ventro-lateral lips of the dermomyotome to the growing limb buds, after which those progenitors start expressing *Pax7* and downregulate the expression of *Pax3* [17,18]. At the epaxial trunk levels, muscles are formed by a de-epithelialization of PAX3+/MYF5+ progenitors within the dermomyotome to form the primary myotome. Further on, a population of PAX3+/PAX7+ progenitor cells located in the central region of dermomyotome will provide the pool of myogenic progenitor cells required for embryonic and fetal epaxial muscles [19–21].

Most head muscles are derived from unsegmented cranial paraxial mesoderm, which comprises cardiopharyngeal mesoderm and prechordal mesoderm, whereas the tongue and subsets of neck muscles are derived from the anterior-most occipital somites [1,22,23]. The formation of the head and the esophagus muscles depends on myogenic progenitors where *Myf5* expression is under the control of the *Tbx1/Pitx2/Isl1* genetic cascades [22,24–30]. Notably, mutant *Pitx2* embryos fail to develop extraocular muscles (EOM) [25,30]. *Pax3* is not expressed in craniofacial myogenic progenitors and *Pax7* is only expressed after the MRFs [31,32]. Skeletal muscles of the neck have either epaxial, hypaxial or cardiopharyngeal mesoderm origin [28,33]. Furthermore, when *Six1* and its co-factor *Eya1* were KO, E17.5 *Six1*$^{-/-}$*Eya1*$^{-/-}$ dKO fetuses exhibited hypoplasia of EOM and branchiomeric skeletal muscles [34].

SIX family proteins are transcription factors characterized by the presence of two conserved domains, a SIX-type homeodomain (HD) that binds to MEF3 elements on the DNA, and an amino-terminal SIX domain (SD) that interacts with co-factors to activate target genes expression including *Pax3*, *MyoD*, *Mrf4*, and *Myogenin* [7,35]. While *Six4*$^{-/-}$[36], *Six2*$^{-/-}$ [37] and *Six5*$^{-/-}$[38,39] fetuses show no major muscle developmental defects, *Six1*$^{-/-}$ fetuses die at birth and exhibit selective limb muscle hypoplasia and lack of diaphragm [15,40]. In *Six1:Six4* double mutants (*s1s4*dKO) epaxial and craniofacial myogenesis still take place, while the hypaxial

phenotype is aggravated by the loss of hypaxial c-Met expression, of all hypaxial musculature and of an impaired expression of *Myogenin*, *MyoD* and *Mrf4* within the myotome [16]. We also previously showed that *Six1* and *Six4* are not crucial for the emergence of PAX7+ myogenic stem cells [41], but necessary for their proper homing at the end of fetal development [42]. Whereas little is known about the implication of *Six2* during embryonic myogenesis, its KO leads to embryonic lethality due to kidney formation defects [37]. *Six2* is expressed in the branchial arches and at the dorsal level of the newly formed somites. It can also bind the *MyoD* MEF3 elements and can thus compensate for the loss of *Six1* and *Six4* in the double KO embryos regarding *MyoD* expression [43]. Last, SIX5 protein is detected in myogenic cells during embryogenesis and in adult myogenic stem cells [44–46].

To determine the specific and overlapping function of SIX homeoproteins during myogenesis at the epaxial, craniofacial, and hypaxial levels and to better characterize their role in the acquisition of the myogenic fate in mesodermal cells within the somites and the branchial arches, we investigated the phenotype of compound *Six* mutants during mouse embryogenesis. We found that *Six1* and *Six2* are required for craniofacial myogenesis, whereas *Six1* and *Six4* are essential at the hypaxial level to activate the myogenic fate of PAX3+ progenitors. Moreover, we showed that even in the absence of all SIX homeoproteins in quadruple mutant fetuses (qKO), epaxial and neck myogenesis, even though highly impaired, still took place with a complete loss of the myogenic PAX7+ cell population by the end of fetal development. Those findings demonstrate that SIX homeoproteins are required for the maintenance of the progenitor PAX7+ cells, but not essential for the onset of primary myogenesis at the epaxial and neck levels. They also demonstrate a combinatorial functional crosstalk between these transcription factors in distinct myogenic territories.

## Results

### *Six1* and *Six4* are required for the myogenic potential of hypaxial dermomyotomal cells of the somite

Myogenic precursors start migrating from the ventral lip of the dermomyotome to the growing limb buds from around E9.5, and express *Pax3* [47]. Those precursors are SIX1+ as well [15] (S1A Fig). We previously showed that in *s1s4*dKO fetuses, hypaxial dermomyotomal cells of E10.5 embryos expressing β-Gal (Six1-β-Gal+ cells, due to the $Six1^{nLacZ}$ allele, have activated *Six1* expression) were misrouted ventrally under the neural tube instead of migrating laterally in the hindlimb bud, and failed to express the MRFs [16]. It has been shown that hypaxial dermomyotomal cells can give rise to endothelial progenitors, smooth, and skeletal muscle progenitors [48,49]. To determine if the absence of SIX1 and SIX4 affects the fate of those precursor cells, we further analyzed the somitic hypaxial Six1-β-Gal+ cells' behavior in the *s1s4*dKO embryos at E10.5 at the hindlimb level. We found that some of them expressed the endothelial marker CD31; others expressed α-SMA (Fig 1A), a marker of the smooth muscle fate, and that they were located in the periphery of the dorsal aorta compared to the heterozygous control cells that migrated towards the hindlimb bud (Fig 1A and 1B).

The equilibrium between *Pax3* and *Foxc1/2* gene expression [50] has also been shown to impact the fate of the PAX3+ myogenic progenitors and their migration to the limb buds. In order to identify a potential disruption in *Foxc1* expression in *s1s4*dKO embryos, we immunostained E10.5 embryo sections at the hindlimb bud level and observed that all the β-Gal + cells in *s1s4*dKO that did not migrate to the limb buds expressed FOXC1 compared to the heterozygous control embryos where no β-Gal+ cells migrating to the limb bud were FOXC1+ (Fig 1C).

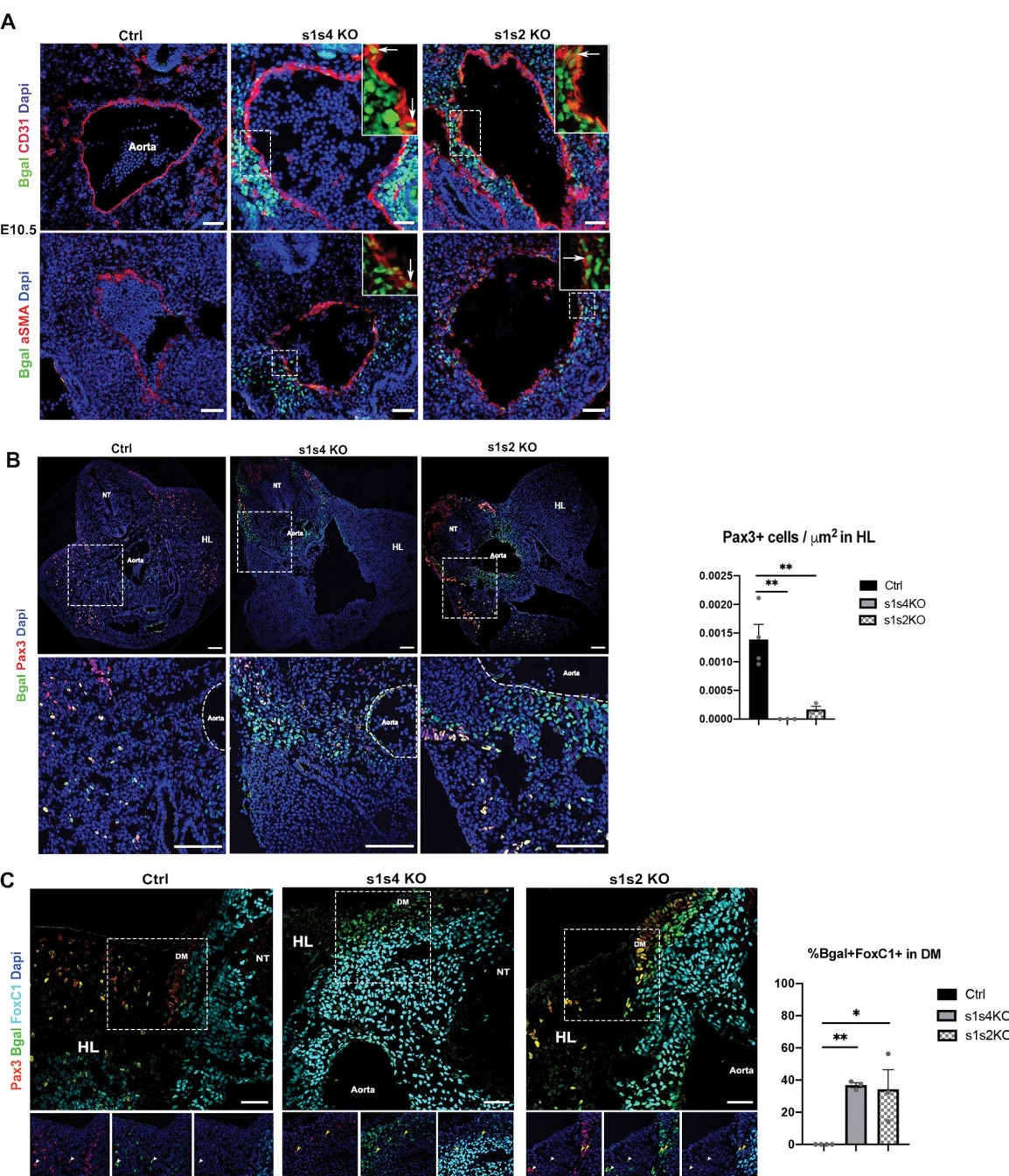

**Fig 1. *Six1*, *Six4* and *Six2*'s requirement in the hypaxial dermomyotomal cells of the somite for limb myogenesis. (A)** Immunostainings on E10.5 Ctrl (s1 Hz), s1s4dKO (n = 3) and s1s2dKO (n = 3) embryos transverse sections at the limb buds level for ß-gal (green), CD31 and α-SMA (red) and Dapi (blue) showing the migration of the ß-gal+ cells under the dorsal aorta in the mutants; Sb = 50μm. **(B)** Immunostaining on E10.5 Ctrl (s1 Hz), s1s4dKO (n = 3) and s1s2dKO (n = 3) embryos transverse sections at the limb buds level for ß-gal (green), Pax3 (red) and Dapi (blue); upper panel zoom shown on the lower panel; NT: Neural Tube, HL: Hindlimb; Sb = 100μm. Right: quantification of the number of PAX3+ cells/μm$^2$ in the hindlimbs of s1s4dKO and s1s2dKO; one-way ANOVA statistical test with mean±s.e.m and ** p<0.005 **(C)** Immunostaining on E10.5 Ctrl (s1 Hz), s1s4dKO (n = 3) and s1s2dKO (n = 3) embryos transverse sections at the limb buds level for ß-gal (green), Pax3 (red), FoxC1 (cyan) and Dapi (blue). Lower panels represent a zoom of the white dashed squares. DM: dermomyotome, HL: hindlimb, NT: neural tube; white arrowheads show Pax3+ ß-gal+FoxC1- cells and yellow arrowheads show Pax3+ ß-gal+FoxC1+ cells; Sb = 50μm. Right: quantification of the percentage of Bgal+FoxC1+ cells in the dermomyotome of s1s4dKO and s1s2dKO; one-way ANOVA statistical test with mean±s.e.m and *p<0.05, **p<0.005.

The decreased expression of *Pax3* in hypaxial dermomyotomal cells of *s1s4*dKO was reported previously [16], and ChIP experiments showed efficient SIX1 [51] and SIX4 binding on the hypaxial enhancer of *Pax3* (S2A Fig). By GMSA we characterized two MEF3 sites in the 291bp hypaxial *Pax3* enhancer located 8kb upstream of the transcription start site (TSS) [52]: efficient binding of SIX proteins to these MEF3 sites was observed (S2C Fig), arguing for a direct control of hypaxial *Pax3* expression by SIX transcription complexes [7].

Altogether, these results demonstrate that *Six1* and *Six4* are necessary for the hypaxial limb myogenesis to occur, by regulating *Pax3* expression in the progenitor cells and maintaining their myogenic fate potential. Decreasing the amount of SIX proteins below a certain threshold in hypaxial dermomyotomal cells leads to their inability to adopt a myogenic fate, adopting an endothelial or smooth muscle fate instead (Fig 1).

## Absence of *Six2* expression worsens hypaxial limb myogenesis defects of *Six1* mutant embryos

Analysis of myogenesis in *Six2* mutants has not yet been reported, though its expression is detected in the somitic dermomyotomes, but not in their hypaxial domains in E9-E10.5 embryos [43]. We show that SIX2 is detected in few migrating PAX3+ cells present in the limb buds at E11.5, while it is mainly detected in the limb bud mesenchyme as already documented ([53], S1A Fig). We observed no muscle malformation in the limbs of E18.5 *s2*KO fetuses (S1B Fig), suggesting that *Six2* is dispensable for migrating hypaxial myogenesis. To test for potential redundant roles of *Six1* and *Six2* in somitic myogenesis we analyzed *s1s2*dKO mutant embryos. Contrary to the absence of PAX3+ cells observed in hypaxial dermomyotomal cells of *s1s4*dKO embryos, PAX3+ hypaxial dermomyotomal cells were detected in *s1s2*dKO embryos and we detected a significant decrease in the number of those cells in the hindlimb of *s1s2*dKO (Fig 1B and 1C). Few Six1-β-Gal+ *s1s2* mutant but not *s1* mutant cells were detected at the dorsal aorta level, expressing α-SMA or CD31 (Figs 1A, S1B and S1C). In order to identify a potential disruption in *Foxc1* or *Pax3* expression in *s1s2*dKO embryos, we immunostained E10.5 embryo sections at the hindlimb bud level. As compared with *s1* mutant embryos, an increased FOXC1 expression was detected in mutant β-Gal+ hypaxial cells of the dermomyotome of E10.5 *s1s2*dKO at the hindlimb level where a few of those β-Gal+ *s1s2*dKO cells expressed FOXC1 (Figs 1C and S1C). Interestingly, similar to *s1s4*dKO, no muscles were formed in the forelimbs of *s1s2*dKO fetuses. While proximal biceps and triceps were reduced in size in single *Six1* mutant, and remaining hypotrophic ventral muscles were detected at the distal level, no major phenotype was observed in single *s2* mutant (S3A and S3B Fig) [15]. Severely reduced muscle masses remained in the distal part of the hindlimb in *s1s2* E18.5 mutant fetuses (S3C Fig).

Contrary to *s1s4*dKO mice lacking all hypaxial musculature including trunk and abdominal muscles, non-migrating hypaxial myogenesis (ventral trunk muscles) is not impaired in *s1s2*dKO (S3B Fig) and the non-migrating hypaxial musculature is well formed. These observations point to unique roles for the different SIX proteins in axial and migrating somitic muscle progenitors.

## *Six1* and *Six2* are required for craniofacial myogenesis

*Six2* is expressed in head mesoderm and in the first and second branchial arches (BAs) [53] and we could detect SIX1, SIX2 and PITX2 in PAX7+ myogenic stem cells and myofibers (MF20+) present at the EOM level in E18.5 fetuses (S4A Fig). Since no major defects were observed in craniofacial myogenesis in *s1*KO and *s1s4*dKO embryos [15,16], we wondered if *Six2* or *Six5* might play a role at this level. We observed that craniofacial muscle masses were

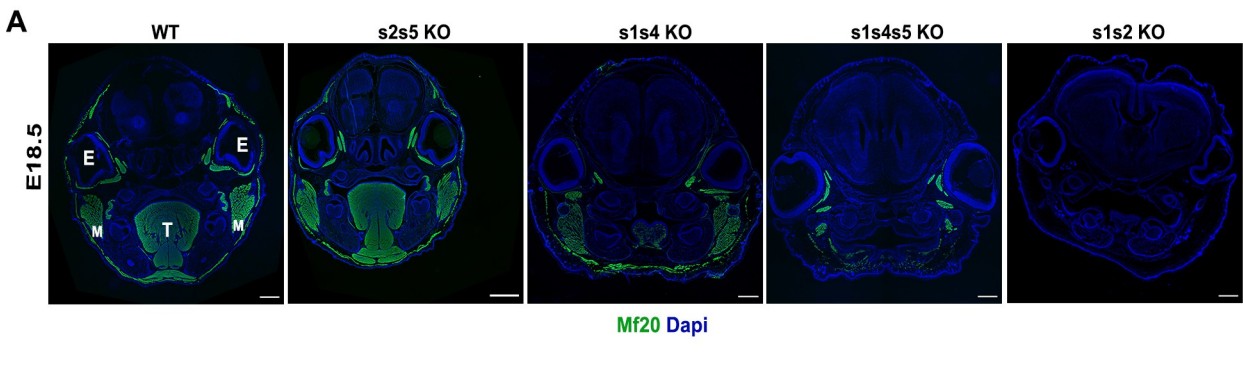

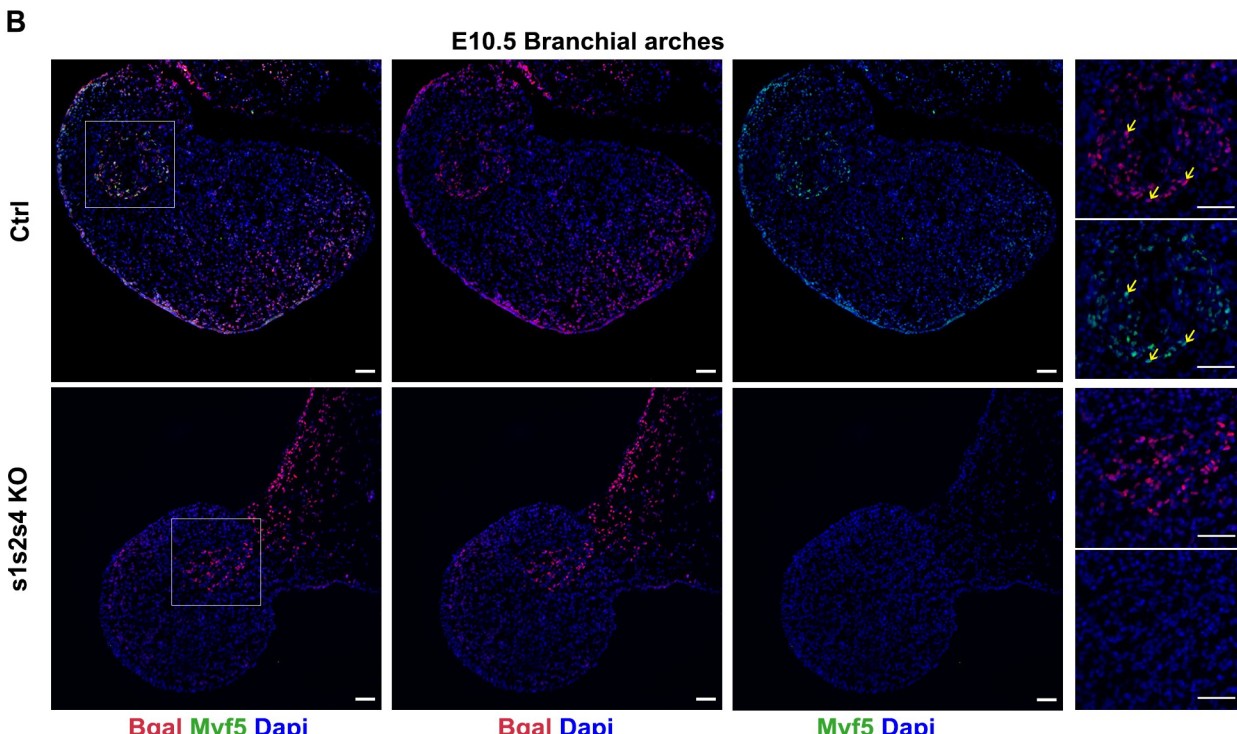

**Fig 2. *Six1* and *Six2*'s requirement for craniofacial myogenesis. (A)** Immunostainings on coronal head sections for E18.5 fetuses at the head level for all the sarcomeric myosins marked by MF20 (green), and Dapi (blue); E: Eye, M: Masseter, T: Tongue, Sb = 500μm. **(B)** Immunostainings on E10.5 Ctrl (s1 Hz), s1s2s4tKO (n = 3) embryos transverse sections at the branchial arches level for ß-gal (red), Myf5 (green) and Dapi (blue); Sb = 50μm; yellow arrows show Myf5+ ß-gal+ cells.

properly formed in *Six2Six5*dKO fetuses (*s2s5*dKO) by the end of fetal development (E18.5) (Figs 2A and S4B). We further investigated the involvement of *Six* genes in cranial muscle masses and observed their reduction in triple *s1s4s5*tKO fetuses as compared to *s1s4*dKO, suggesting that *Six5* or the SIX protein dosage may influence craniofacial myogenesis. While the masseters, derived from the first branchial arch, formed normally in *s1s4*dKO, they were hypoplasic in *s1s4s5*tKO (Fig 2A). Furthermore, both *s1s4s5*tKO and *s2s5* dKO muscles masses show the presence of PAX7+ cells (S4B Fig). This demonstrates that in a *s1s4*dKO context, *Six5* is required for craniofacial muscle growth, and that *Six2* alone is not sufficient to sustain efficient muscle development at the head level. Unexpectedly *s1s2*dKO showed a total absence of craniofacial myogenesis (Fig 2A), also observed in *s1s4s2*tKO (S4C Fig), pointing to functional redundancy of *Six1* and *Six2* in the regulation of craniofacial myogenesis.

*Myf5* is the first MRF expressed in the branchial arches (BA) of the mouse embryos. We detected no MYF5+ cells in the BA of the triple *s1s4s2tKO* embryos at E10.5 while β-Gal+ cells were present, at this location (Fig 2B). This data excludes a migration defect of mutant cranial mesodermal cells in the BAs and suggest that mutant mesodermal cells present in the BA have an impaired ability to activate Myf5-dependent skeletal myogenesis in the mutant. The subsequent fate of these β-Gal+ cells remains to be elucidated. Contrary to what was observed at the somitic level, there is no evidence that mesodermal craniofacial *s1s2* mutant precursors cells adopt a smooth muscle fate (S5A and S5B Fig).

The regulatory sequences allowing *Myf5* expression in the BAs have been characterized [4,5]. By GMSA, we detected, the presence of at least one MEF3 site in the *Myf5* BA enhancer that can be recognized by SIX (S2C Fig). Interestingly, using transient transfection Luciferase assays in MEF qKO cells, we found that this enhancer is activated by the expression of SIX proteins (S2D Fig), arguing for a direct role of SIX homeoproteins in the activation of *Myf5* in the BA, which allows myogenic fate acquisition in the mesodermal cells.

Furthermore, SIX1 and SIX4 ChIP-Seq data revealed SIX binding sites both in *Myf5* branchial arch enhancer and in the promoter regions of *Pitx2* [54,55] (S2B and S2E Fig), suggesting that SIX may also regulate this upstream gene as well.

Since embryos lacking *Pitx2* fail to develop EOM and since *s1s2*KO fetuses lack all craniofacial musculature including the EOM, we next examined the expression of PITX2, a known upstream regulator of *Myf5/MyoD* in the EOM [24,25] in E10.5 *s1s4s2tKO* embryos. In the absence of those *Six* genes, PITX2 was still detected at the EOM level although not in all the β-Gal+ cells (Fig 3A), suggesting that SIX are not strictly required for *Pitx2* expression. Those results suggest that in the absence of SIX, nuclear PITX2 proteins cannot promote skeletal myogenesis in EOM. We observed Desmin expression in β-Gal+ cells of *s1s4s2tKO* (Fig 3B).

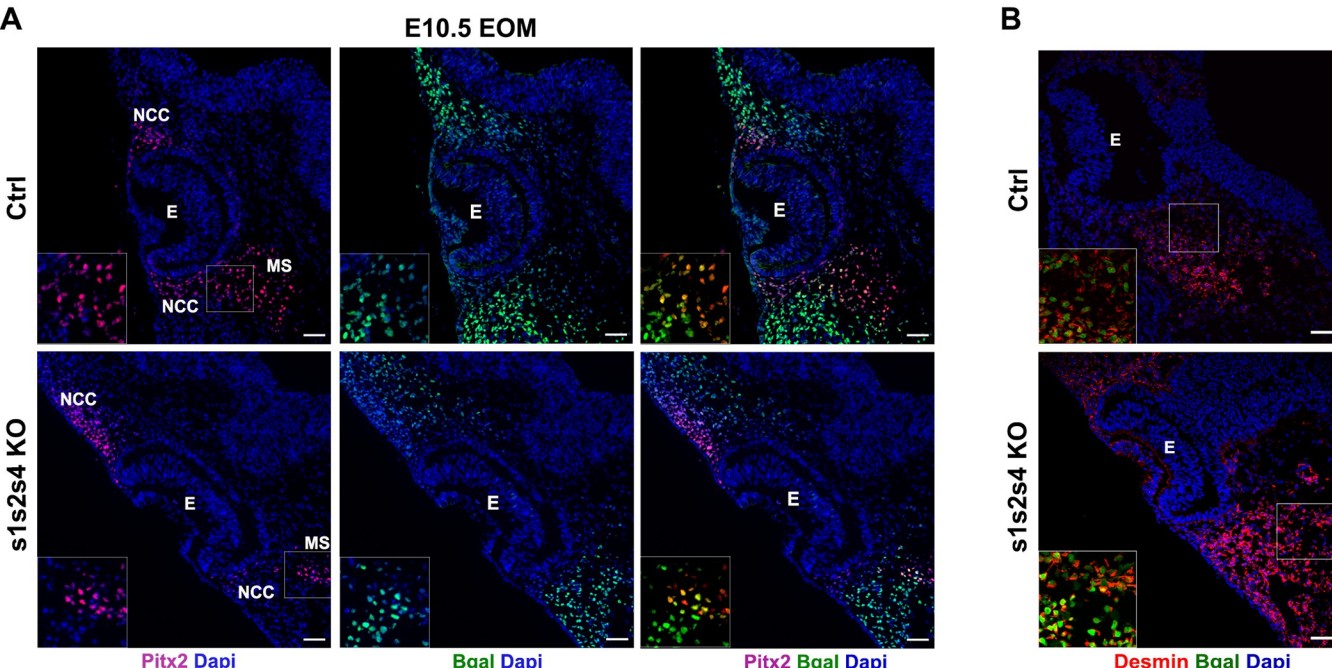

**Fig 3. *Six1 Six2* and *Six4*'s requirement for EOM formation but not for *Pitx2* expression. (A)** Immunostainings on E10.5 Ctrl (s1 Hz), s1s2s4tKO (n = 3) embryos transverse sections at the head level (EOM) for ß-gal (green), Pitx2 (red) and Dapi (blue); NCC: Neural Crest Cells, E: Eye, MS: Mesenchymal cells, Sb = 50μm. **(B)** Immunostainings on E10.5 Ctrl (s1 Hz), s1s2s4tKO (n = 3) embryos transverse sections at the head level (EOM) for ß-gal (green), Desmin (red) and Dapi (blue); E: Eye, Sb = 50μm.

However, we discarded the possibility that mutant progenitor cells may adopt a smooth muscle fate since Calponin staining was not observed at the EOM level in *s1s2* mutant embryos (S5B Fig).

Since *Six1* and *Six2* are also expressed in cranial neural crest cells (CNCC) where they control frontonasal development [56], and an interplay between the development of craniofacial muscles and CNCC orchestrates harmonious craniofacial development [57–61], we analyzed craniofacial myogenesis in *Six2-/-;Six1flox/flox;Wnt1-CRE* embryos, where *Six1* is only deleted in CNCC [56]. While BA-derived muscles formed without major defects, we observed severe EOM hypoplasia in mutant *Six2-/-;Six1flox/flox;Wnt1-CRE* (Fig 4A).

As esophagus striated muscle has also a cranial mesodermal origin [26,29] we investigated its presence in *s1*KO, *s2*KO and *s1s2*dKO fetuses at E18.5. While smooth muscle cells of the esophagus were present in these different mutant fetuses, yet we failed to detect any MF20 + cells at this level in *s1*KO and *s1s2*dKO fetuses, whereas *s2*KO fetuses did not show any defects at this level (Fig 4B). The fact that we also failed to detect β-Gal+ cells in the esophagus of *Six1-/-* fetuses (S6 Fig) demonstrates that *Six1-/-* myogenic progenitors did not migrate from the posterior arches to colonize the mutant esophagus. We previously observed a lack of diaphragm muscle in *Six1-/-* mutants and a lack of *c-Met* expression in hypaxial somitic cells of *Six1Six4*dKO, arguing for a direct control of *c-Met* expression by SIX homeoproteins [15,16]. Accordingly, ChIP-seq experiments [54,55] revealed multiple SIX binding at the *c-Met* locus (S2F Fig). As the MET-HGF pathway is also implicated in migration of esophageal progenitors [29], the lack of esophagus striated muscle in *s1*KO and *s1s2*dKO fetuses is also probably due to a lack of activation of *c-Met* in these mutants.

## SIX homeoproteins are required for the progression of epaxial primary myogenesis and necessary for fetal myogenesis and the maintenance of the precursor cells' pool

We previously showed that the loss of *Six1* and *Six4* expression abolishes hypaxial myogenesis, while epaxial myogenesis, although impaired, still takes place but leads to hypoplasic dorsal muscles at E18.5 [16,42,62]. Epaxial muscle masses in triple E14.5 *s1s2s4*KO and E18.5 *s1s4s5*tKO fetuses have a very reduced number of myofibers compared to the control, but still formed along with their associated PAX7+ cells (S7A Fig). To fully investigate if SIX homeoproteins are required for epaxial myogenesis, we generated quadruple KO fetuses lacking the four *Six* genes expressed in the muscle lineages: *Six1*, *Six2*, *Six4* and *Six5* (*q*KO). Surprisingly, we observed that those fetuses still showed some highly disorganized MYH+ myofibers at E14.5, as revealed by MF20 staining, forming a dorsal muscle mass that was severely reduced in size, as well as some neck muscles (Fig 4C). This residual epaxial muscle mass, however, did not increase in size between E14.5 and E18.5 (Fig 5A), the time by which secondary myogenesis results in the genesis of new secondary myofibers on the scaffold of primary myofibers [63]. These observations showed that SIX homeoproteins are not required for the early steps of epaxial primary myogenesis but are necessary for its progression and mandatory for fetal myogenesis establishment. At the neck level we observed disorganized muscles at E14.5 and E18.5 (Figs 4C and S8) in *q*KO. At E14.5 while some neck muscles in wt embryos show the presence of ISL1/2+ and FOXP2+ cells, suggesting their craniofacial origin [61], neck muscles present in the *q*KO show a reduction or absence of those cells, arguing for their somitic origin. We tentatively identified remaining neck muscles masses as the splenius muscles in *q*KO embryos.

In mammals, primary myofibers express embryonic (fast) and slow-Myosin Heavy Chain (MYH). In contrast, secondary fibers express the fast embryonic and perinatal isoforms from their inception and not slow MYH [64]. Given that *Six*1 and *Six*4 were shown to be necessary for the induction of the fast-type muscle program during embryogenesis [62,65], we wanted to

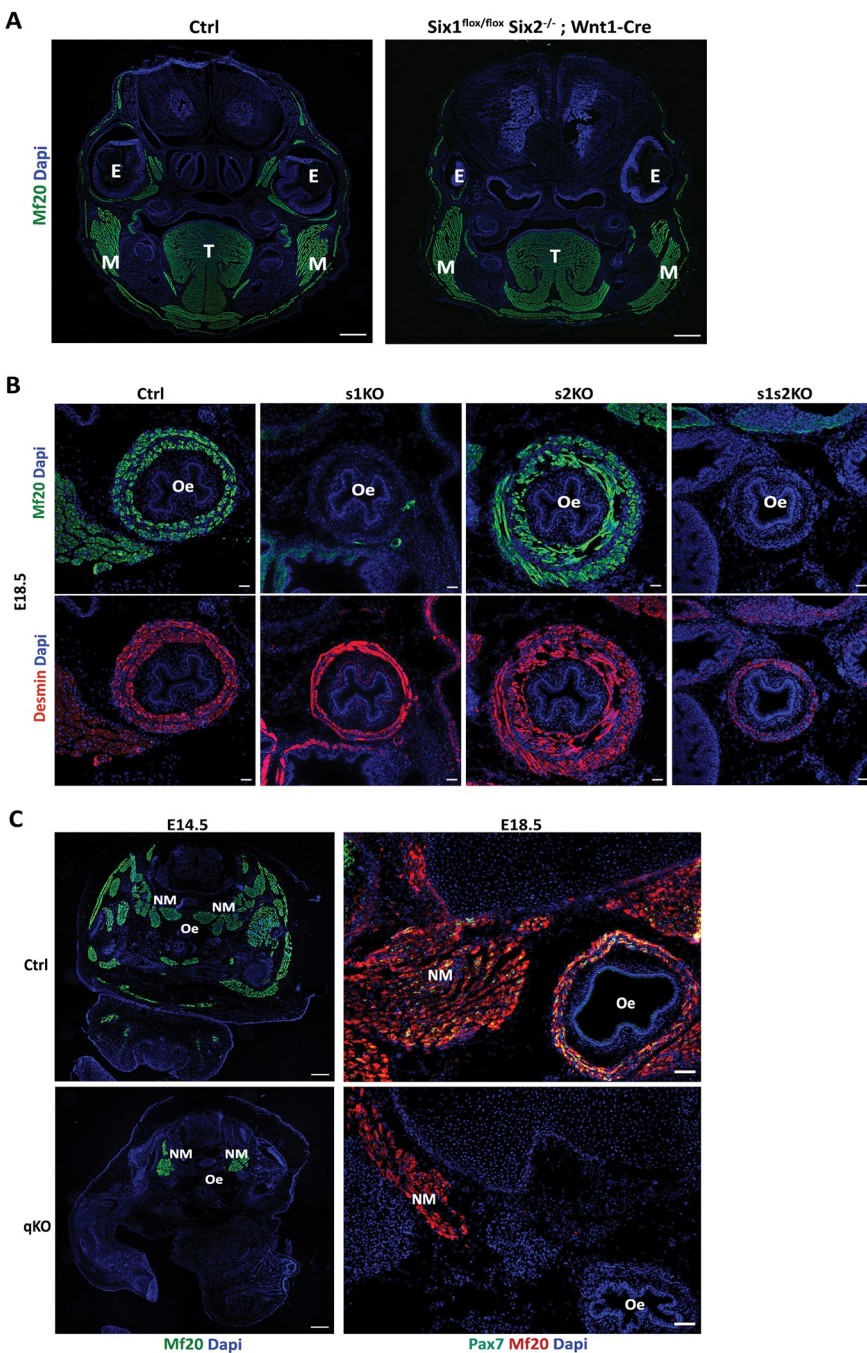

**Fig 4. *Six1* and *Six2*'s requirement for cranial-mesoderm-derived esophagus musculature. (A)** Immunostaining on head coronal section of E16.5 Ctrl (n = 3) and Wnt1-Cre; Six1$^{Lox/Lox}$ Six2$^{-/-}$ (n = 3) fetuses showing impairments in EOM muscle formation by MF20 (green), Dapi (Blue); E: Eye, T: Tongue, M: Masseter, Sb = 200μm. **(B)** Immunostainings on transverse trunk sections for E18.5 Ctrl (s1 Hz), s1KO (n = 3), s2KO (n = 2) and s1s2dKO (n = 3) fetuses for sarcomeric myosins marked by MF20 (green) and the smooth muscle marked by Desmin (red); oe: esophagus, Sb = 30μm. **(C)** Left panel: Immunostainings on transverse trunk sections for E14.5 Ctrl and qKO (n = 3) fetuses showing sarcomeric myosins marked by MF20 (green), and Dapi (blue); NM: Neck Muscles, oe: esophagus, Sb = 300μm. Right panel: Immunostainings on transverse trunk sections for E18.5 Ctrl and qKO (n = 3) fetuses zoomed at the neck level showing sarcomeric myosins marked by MF20 (red), Pax7 (green) and Dapi (blue); NM: Neck Muscles, oe: esophagus, Sb = 150μm.

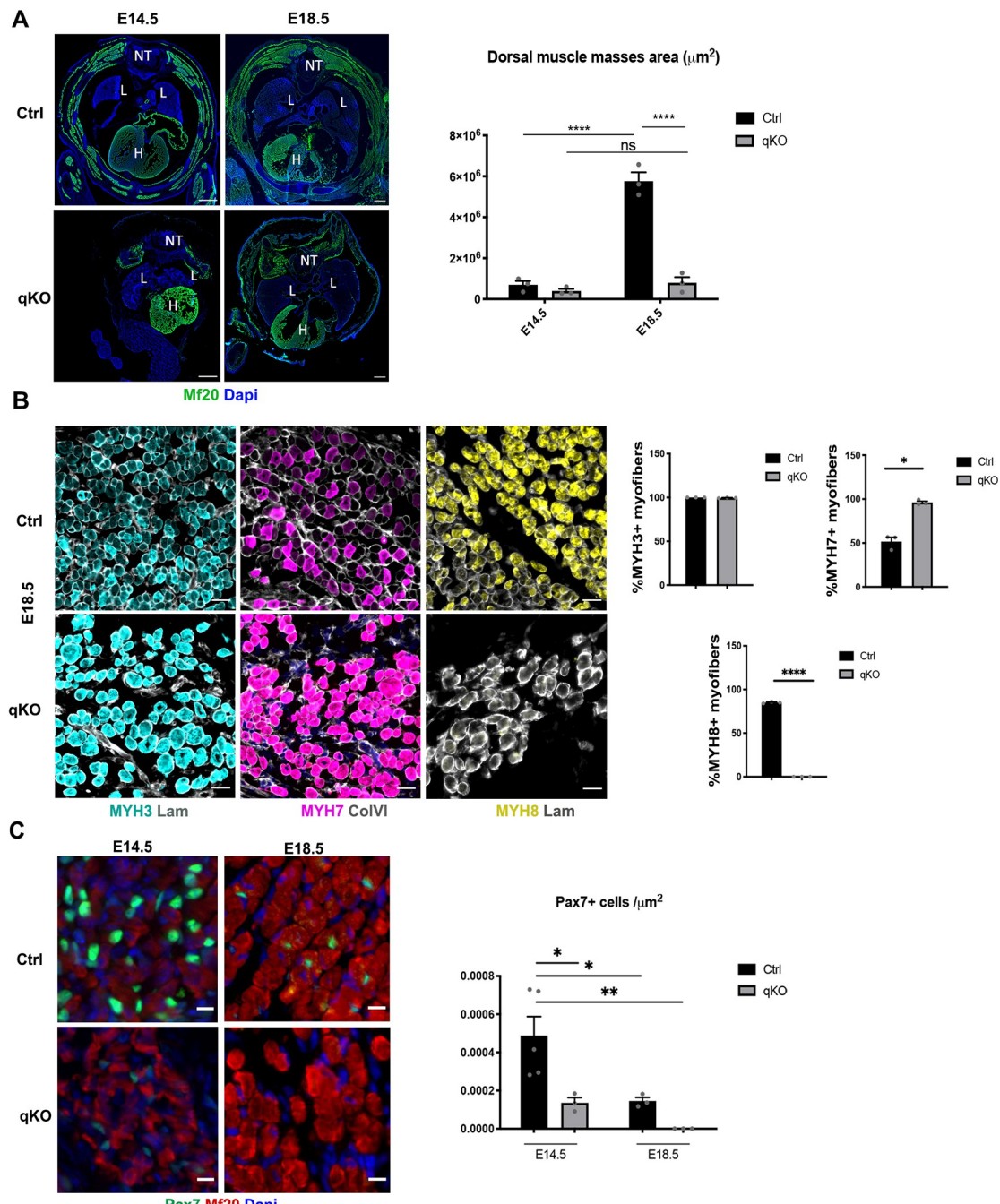

**Fig 5. SIX homeoproteins are required for fetal myogenesis and for the maintenance of the myogenic progenitor's pool. (A)**
Left: Immunostainings on transverse trunk sections of E14.5 and E18.5 Ctrl and qKO (n = 3) fetuses at the trunk level showing sarcomeric myosins marked by MF20 (green) and Dapi (blue); NT: Neural Tube, L: Lungs, H: Heart, Sb = 300μm. Right: Quantification of the MF20+ muscle mass area (μm$^2$) at the dorsal level of E14.5 and E18.5 Ctrl and qKO (n = 3) fetuses; two-way ANOVA statistical test with mean ±s.e.m and **** p<0.0001. **(B)** Left: Immunostainings on transverse trunk sections of E18.5 Ctrl and qKO (n = 3) fetuses showing the different types of myosins in the dorsal muscle masses area; MYH3 (Cyan), MYH7 (Magenta), MYH8 (Yellow), Laminin (Grey), Sb = 20μm. Right: Quantification of the percentage of MYH+ myofibers in Ctrl and qKO dorsal muscle masses; statistical paired t-test with mean ±s.e.m and *p<0.05 **** p<0.0001 **(C)** Left: Immunostainings on transverse trunk sections of E14.5 and E18.5 Ctrl and qKO (n = 3) fetuses at the trunk level showing on the dorsal muscle masses sarcomeric myosins marked by MF20 (red), Pax7 (green) and Dapi (blue); Sb = 10μm. Right: Quantification of the number of Pax7 + cells per μm$^2$ on the dorsal muscle masses of E14.5 and E18.5 Ctrl and qKO (n = 3) fetuses; two-way ANOVA statistical test with mean±s.e.m and * p<0.05 **p<0.005.

confirm the implication of SIX in fetal myogenesis by assessing the expression of MYH isoforms. As suspected, qKO fetuses showed an increase in embryonic MYH (MYH3) as well as slow MYH (MYH7) positive myofibers, while no fast perinatal MYH (MYH8) was detected in their remaining epaxial muscle masses (Fig 5B). This is also in agreement with the activity of a super enhancer at the fast *Myh* locus required for *Myh8* but not *Myh3* expression, and whose activity is modulated by SIX homeoproteins [66].

PAX7+ myogenic cells contribute to the formation and growth of the muscle tissue during the fetal period (increase of myofibers' number until E18.5 –hyperplasia-) and the perinatal period (increase of the myonuclei number in the existing myofibers–hypertrophy-) [42,67–69]. We previously showed that in the absence of SIX1 and SIX4, PAX7+ cells present a homing defect [42]. We therefore asked if the absence of SIX impacted the behavior of PAX7+ cells at E14.5, a period when the majority of epaxial PAX7+ cells expressed SIX2 (S7B Fig). Notably, in the absence of all myogenic *Six* genes, there was a drastic decrease in the number of PAX7 + cells at E14.5 compared to the control; those cells were completely lost by E18.5 (Fig 5C). The same result was also observed in the remaining neck muscle masses (Fig 4C). Accordingly, we observed a decreased percentage of PAX7+ CyclinD1+ cells in the qKO E14.5 fetuses in epaxial muscles (S7C Fig). This decrease in number is accompanied by a decrease in PAX7 nuclear accumulation since its detection starting at E14.5 was only possible after a drastic antigen retrieval treatment not necessary for control sections (refer to Materials and Methods). These results suggest that the lower level of PAX7 in myogenic progenitors may be the consequence of decreased *Pax7* gene expression, due to an impaired SIX protein binding to important enhancer elements of *Pax7* involved in its expression. To identify enhancer elements at the *Pax7* locus that may be controlled by SIX, we used snATAC-seq data of *Pax7*-expressing quiescent satellite cells (QSC) of adult muscles [70]. Furthermore, DNA regions of *Pax7* present in BAC transgenic mice showing a correct spatiotemporal expression of *Pax7* restrict the analysis of enhancer elements controlling its expression [71,72]. In addition to the promoter region, we observed several opened DNA regions at the *Pax7* locus in QSC, as compared with the other cell types present in adult muscles such as fibro adipogenic progenitors (FAPS); the +60kb and +140kb DNA regions are open exclusively in QSC (S9A Fig). We further compared those open chromatin regions at the *Pax7* locus to regions with evolutionary conserved MEF3 sites present in the vicinity of MYOD binding sites [73], and to known SIX1 and SIX4 binding sites from ChIP-seq in proliferating myogenic cells [54,55]. Interestingly three DNA regions at +40 kb, +60kb and +140kb at the *Pax7* locus are open in QSC and contain MEF3/MYOD binding sites (S9B Fig). The +60kb and the +140kb region establish looping with the promoter region (Zhao et al, bioRxiv 2021.12.20.473464; https://doi.org/10.1101/2021.12.20.473464), with the +140kb region being absent from the BAC used for Pax7-GFP transgenic animals ([71,72], (Foteini Mourkioti, U. Pennsylvania, personal communication). The +60kb is required for *Pax7* expression in induced ESC and corresponds to the En7 element described in [74]. We identified several MEF3 sites in the +60kb and +140kb region able to bind recombinant SIX proteins by GMSA (S9C and S2C Figs). Furthermore, transient transfection assays of Luciferase reporters under the control of the +60 and the +40 Pax7 enhancers in C2C12 myoblasts showed their efficient activation by the constitutively active SIX1-VP16 chimeric protein (S9D Fig), confirming the ability of SIX proteins to act on the identified *Pax7* enhancer elements. Those results suggest that SIX homeoproteins control the maintenance of the myogenic progenitor cells pool at least by directly controlling *Pax7* gene expression. We cannot exclude the possibility that early *Pax7* expression observed in *Six*qKO embryos does not depend on the MEF3 sites that we identified in enhancer elements of *Pax7*, but it is possible that these MEF3 sites, in the absence of SIX proteins, are recognized by other transcription factors. In favor of this hypothesis we showed that DNA elements bound by NKX2.5 in the *Rspo3* gene [75] or

HOX homeoproteins in the *Mlc* gene [76] can be bound by SIX in GMSA (S9C Fig), suggesting that NKX2.5, SIX and HOX may bind common DNA elements in the genome.

## SIX homeoproteins are required for the maintenance of the precursor-cell state of PAX7+ cells

We hypothesized that PAX7+ cells are exhausted at the end of fetal development due to their lower PAX7 protein content and their premature differentiation. Interestingly, we observed a significantly higher percentage of PAX7+MYOD+ cells at E14.5 in the qKO embryos compared to the control (Fig 6A). To determine if they eventually engage in the differentiation program, we examined the expression of the differentiation marker MYOG and found that, while in the control at E14.5 the majority of the PAX7+ cells were MYOG-Ki67+ (69.8%), in the qKO embryos this subpopulation of PAX7+ cells was significantly lower (42%) and equally divided with the MYOG+Ki67+ subpopulation (42% and 41.6% respectively). This result, together with the higher percentage of PAX7+MYOG+Ki67- cells in the qKO (5.68%) compared to the control (0.36%), made the proportion of PAX7+ cells undergoing differentiation higher in the qKO embryos, and thus lead to their exhaustion (Fig 6B).

## Discussion

The aim of this study was to decipher the selective role of different SIX homeoproteins in the control of myogenic cell behaviors during embryogenesis at different body levels. We examined the muscle phenotype of compound *Six* mouse mutant embryos and showed that different sets of *Six* genes are critical to allow multipotent embryonic progenitors to adopt a myogenic fate in distinct anatomical locations. We further show that in the absence of the four *Six* genes expressed in myogenic cells, fetal myogenesis is selectively impaired due to the loss of PAX7+ myogenic stem cells that differentiate prematurely. A schematic representation of our findings as well as a table resuming the myogenic consequences of *Six* genes deletion are illustrated in Figs 7 and S10.

### *Six1* and *Six4* play a role in myogenic commitment at the hypaxial level

*Six1* and *Six4*, two genes expressed in hypaxial dermomyotomal cells, are involved in maintaining *Pax3* expression by binding to its hypaxial enhancer and allowing lineage progression of the precursor cells, as well as their migration to the limb buds. *s1s4*dKO thus, leads to a total absence of limb musculature [51]. ChIP experiments showed binding of SIX1 on the hypaxial *Pax3* enhancer in mouse embryos [51], and we identified two MEF3 sites bound by SIX in this DNA element that may be responsible for the SIX activity-dependent hypaxial *Pax3* enhancer activity. The decreased expression of PAX3 in hypaxial progenitor cells in *s1s4*dKO and the observed upregulation of *FoxC1* would explain their change in fate towards smooth muscle. In fact, it has been shown that hypaxial dermomyotomal cells of the somites can temporally give rise to endothelial, smooth and skeletal muscle progenitors [49] where the balance between non-canonical Wnt [13], Notch and BMP signaling [48,77] modulates the behavior of these cells and the equilibrium between *Pax3* and *Foxc1/2* gene expression [50]. The latter study also revealed a negative feedback loop between *Pax3* and *FoxC1/2* with *Pax3* expression favoring a myogenic fate acquisition at the expense of non-myogenic fate driven by *FoxC1/2*. Here, we also showed that, while *Six1* together with *Six4* are necessary for both, the non-migratory and migratory hypaxial myogenesis, ventral thoracic and abdominal non-migrating musculature is present at E18.5 in *s1s2*dKO fetuses and that some PAX3+ cells of their hypaxial somitic dermomyotome at the limbs level are able to acquire a myogenic identity and migrate into the limb bud to form residual MF20+ fibers. We also observed a severe aggravation of the muscle

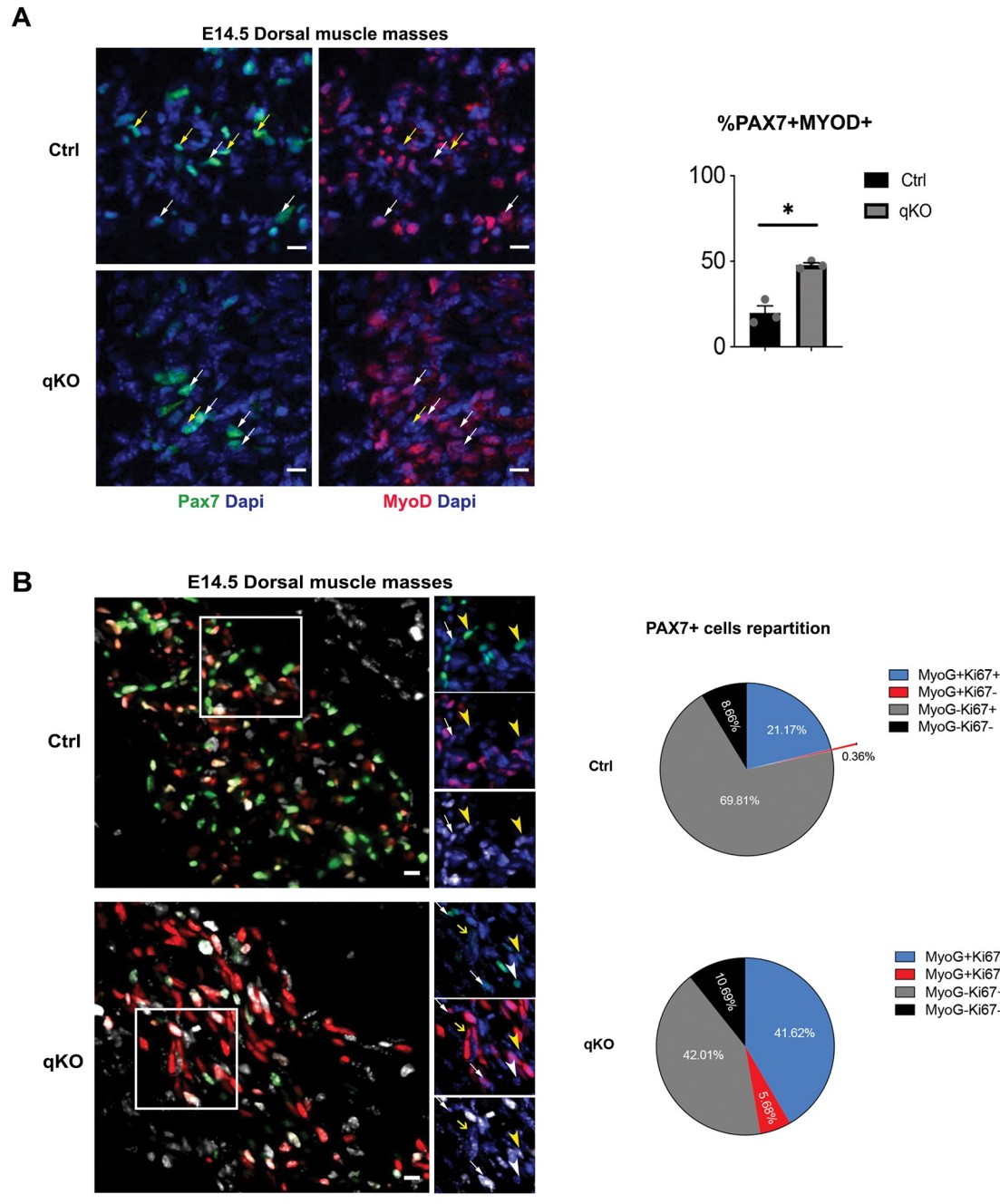

**Fig 6. SIX homeoproteins are required for the maintenance of the precursor-cell state of PAX7+ cells. (A)** Left: Immunostainings on transverse trunk sections of E14.5 Ctrl and qKO (n = 3) fetuses at the trunk level showing on the dorsal muscle masses Pax7 (green), MyoD (red) and Dapi (blue); white arrows indicate Pax7+MyoD+ cells, yellow arrows indicate Pax7 +MyoD- cells, Sb = 10μm. Right: Quantification of the percentage of Pax7+MyoD+ cells in the dorsal muscle masses of Ctrl and qKO fetuses at E14.5; Welch's test done with mean ±s.e.m and * p = 0.0141 **(B)** Left: Immunostainings on transverse trunk sections of E14.5 Ctrl and qKO (n = 3) fetuses at the trunk level showing on the dorsal muscle masses Pax7 (green), Myog (red), Ki67 (grey) and Dapi (blue); Sb = 10μm. Right: Quantification of the percentage of the different subpopulations of Pax7+ cells in the dorsal muscle masses of Ctrl and qKO fetuses at E14.5: in the control the majority of the PAX7+ cells are MYOG-Ki67+ (69.81%), the rest are divided between MYOG+Ki67+ (21.17%) and MYOG-Ki67- (8.66%) and very few (0.36%) are MYOG +Ki67-; in the qKO fetuses the majority of Pax7+ cells are equally divided between MYOG-Ki67+ (42.01%) and MYOG+Ki67+ (41.62%) and the rest is MYOG-Ki67- (10.69%) and MYOG+Ki67- (5.68%). Statistical two-way ANOVA test %Pax7 +MYOG-Ki67+ is significantly lower in the qKO fetuses with mean ±s.e.m and * p = 0.0216.

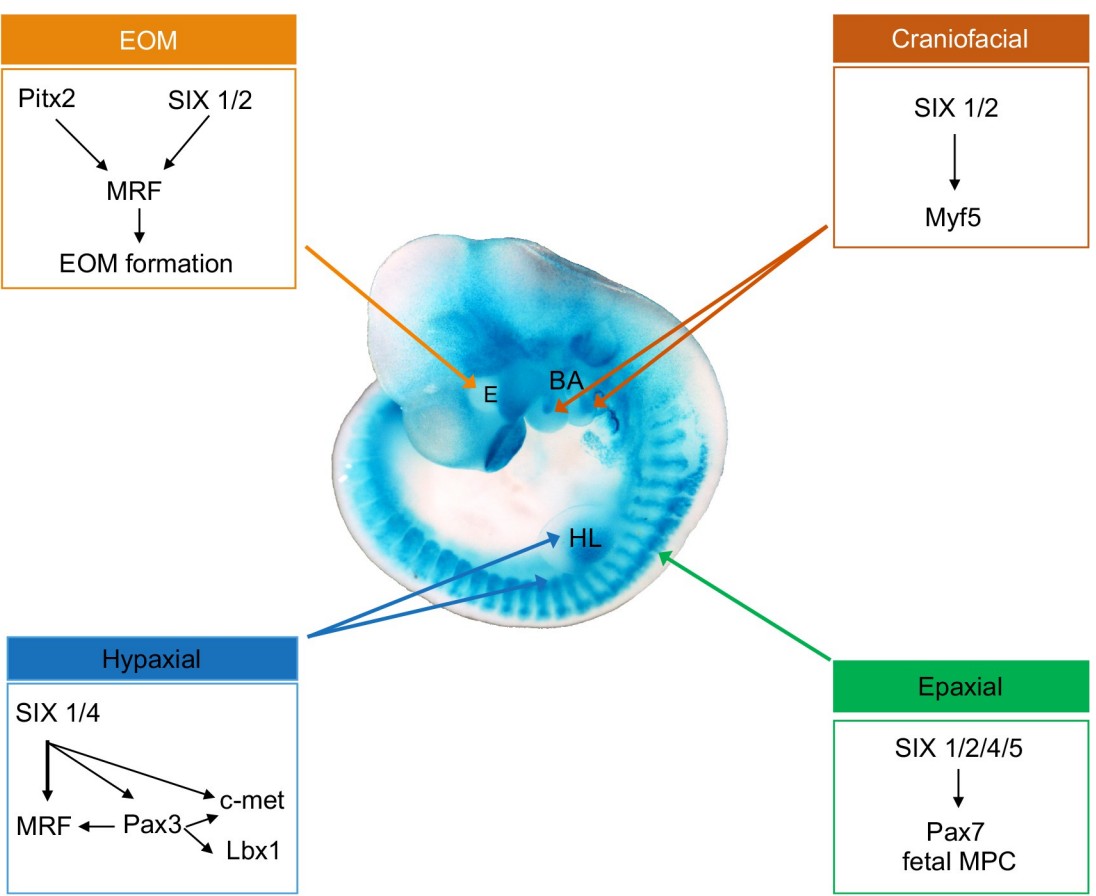

**Fig 7. Schematic representation illustrating a ß-gal stained $Six1^{nLacZ/+}$ embryo of E10.5 and the control of myogenesis by SIX homeoproteins on different embryonic territories, as confirmed by the findings of this study.** E: eye, BA, branchial arches, HL, hindlimb.

limb phenotype in *s1s2*dKO as compared with the *s1* mutant phenotype [6,15,40]. This aggravation may be linked with the expression of *Six2* in myogenic stem cells and/or in surrounding mesenchymal cells of the limb buds [53].

## *Six1* and *Six2* are necessary for *Myf5* expression and craniofacial myogenesis

In this study, we further show that *s1s2*dKO completely abrogates craniofacial myogenesis. Further, we showed that at the craniofacial level the combined action of *Six1* and *Six2* is required in branchial arch mesodermal progenitors to activate *Myf5* and allow the commitment of those cells to the myogenic lineage. The presence of *Six1* and *Six2* in the craniofacial mesoderm [53,78,79] suggests that the absence of *Myf5* activation observed in mutant BAs may be a cell autonomous process. Nevertheless, *Six1* and *Six2* are also expressed in craniofacial neural crest cells (CNCC) [56] and $Six2^{-/-}$, $Six1^{-/-}$ and $Six2^{-/-}$:$Six1^{flox/flox}$:*Wnt1-CRE* mutant fetuses show severe skull/mandibular abnormalities [56,78–80]. Excess BMP and endothelin signaling may promote severe craniofacial malformations (frontonasal, maxillary and mandibular) in *s1s2*dKO fetuses, since it was shown that reducing BMP4 or EDNRA signaling in *s1s2*dKO reduced the penetrance of maxillary malformations [56,80,81]. Whether increased BMP signaling from CNCC in $Six2^{-/-}$:$Six1^{flox/flox}$:*Wnt1-CRE* embryos is responsible for EOM

hypoplasia is an interesting possibility given that increased BMP signaling decreases *Myf5* expression in EOM progenitors [82]. As such, excessive BMP signaling in *s1s2*dKO embryos may thus impair the myogenic fate acquisition of EOM mesodermal progenitors.

Like the dermomyotomal cells are able to give rise to distinct cell fates, bipotent meso-dermal cranial cells have been characterized with *Foxp2* being among the key drivers of the non-myogenic fate [61]. Those bipotent cranial mesodermal cells express *Myf5* and *Six1*, and one can suspect that the absence of *Six1* impairs *Myf5* expression, thereby impairing their pro-gression toward the myogenic fate, as observed in *s1s4*dKO embryos at the hypaxial level. The fact that the initiation of *Myf5* expression in BA and EOM still takes place after ablation of CNCC in chick embryos [57,82] let us speculate the growth impairment of EOM in mutant *Six2*<sup>-/-</sup>:*Six1*<sup>flox/flox</sup>:*Wnt1-CRE* fetuses may be due to the presence of a complex interplay between mutant CNCC overexpressing BMP signaling and *Six2*<sup>-/-</sup> EOM myogenic progenitors.

*Pitx2* is still expressed in CNCC and mesoderm of *s1s2*dKO embryos at the EOM level but is not sufficient to activate the myogenic program in mutant mesodermal cells at the eye level, showing the absolute requirement of one of these two *Six* genes to activate *Myf5*/*MyoD*.

In BA1, *Pitx2* acts upstream of *Six2* and *Myf5*, and BA1 myogenesis is aborted in *Pitx2* mutants, while absence of *Pitx2* in BA2 does not impair myogenesis [24,83]. We showed that *s1s2s4*tKO craniofacial mesodermal cells can migrate to the BAs and are Ki67<sup>+</sup>. However, we failed to detect MYF5 in the β-Gal+ cells present in the BA at E10.5. *Pitx2*, *Tbx1*, *Capsulin* and *MyoR* are required for branchiomeric muscles formation upstream of *Myf5* [23,84–86]; whether the expression of those genes is modulated by *Six1* and *Six2* at the BA level remains to be determined.

While craniofacial muscles do form in *Six1*<sup>-/-</sup> embryos [15], we showed that such mutants are devoid of esophagus skeletal muscles and of migrating β-Gal+ progenitors that have a cra-niofacial origin. Interestingly, those progenitors must activate *c-met* expression, allowing their migration to the esophagus anlagen [29]. *s1s4*dKO embryos are deficient for *c-met* expression [15,16] and since ChIP-seq experiments show SIX1 protein binding the promoter of *c-met* (S2F Fig), we suggest that SIX1 is required in craniofacial myogenic progenitors to activate *c-met* to allow their migration to colonize the esophagus.

Altogether, in contrast to the *Pax/Six* network described in the somites, our data uncovered an unsuspected role of *Six1* and *Six2* as major upstream regulators of craniofacial myogenesis, independent of *Pax3*. This result is reminiscent of the requirement of *Six1a* in zebrafish for the genesis of EOM [87], and of the abrogation of Amphioxus head muscle formation after injec-tion of transdominant negative *Six1/2-Engrailed* mRNA in embryos [88]. As myogenic stem cells show differential properties depending on their embryonic origin [89–91] the identifica-tion of *Six2* as a major craniofacial myogenic regulator upstream of *Myf5* suggests that it may participate with well-known transcription factors governing craniofacial myogenesis (*Tbx1*, *TCF21/Capsulin*, *Pitx2*, *MyoR/Musculin*, *Six1*) to confer specific properties to these myogenic cells and their corresponding muscles.

## Epaxial fetal myogenesis and PAX7+ cells' pool maintenance depends on *Six*

The phenotype we describe in qKO fetuses, that develop some reduced epaxial and neck mus-culature, indicate that the presence of SIX1, SIX2, SIX4 and SIX5 is not required for the onset of primary epaxial myogenesis, but necessary for its progression and mandatory for fetal myo-genesis. This result is similar to the phenotype of *Pax3*<sup>-/-</sup>*Pax7*<sup>-/-</sup> embryos in which a primary myotome also formed [20]. This shows that a *Pax/Six*-independent primary myogenesis can

be initiated at the axial level in the embryo but is arrested before birth. We show in the present study that specific MEF3 sites bound by SIX homeoproteins can also be recognized by HOX and NKX2.5 proteins. Whether the *Pax/Six*-independent early myogenesis observed in *Pax3/ Pax7* or *Six1/Six2/Six4/Six5* mutant embryos is under the control of homeoproteins that bind some of the cis-regulatory DNA elements recognized by SIX is an intriguing possibility.

In the absence of the four myogenic *Six* genes [7,45,92,93], the PAX7+ progenitor cell population is not maintained. Instead, PAX7+ cells express *Myog* and differentiate prematurely, leading to their complete exhaustion by the end of fetal development (E18.5) and to an absence of epaxial muscle growth after E14.5. We and others have shown that MYOD binding was severely impaired in cells devoid of SIX activity [73,93]. Moreover, MYOD ChIP-seq data [94] show co-binding with SIX1 in the +40 kb region, and together with SIX4 in the +140kb region of *Pax7*. Because MYOD is required to activate *Pax7* during embryonic development to keep the myogenic identity [95], it is thus possible that decreased MYOD binding at the *Pax7* locus in *Six*qKO may also participate to the downregulation of *Pax7* in those mutant embryos.

As PAX7 expression is low in qKO myogenic cells, it is not sufficient to allow those cells to self-renew, demonstrating that SIX proteins are required for fine-tuning the balance between self-renewal and differentiation. Those two processes require SIX proteins, and their absence leads to the concomitant expression of PAX7 and MYOG in the precursor cells. Nevertheless, these findings do not exclude the possibility that SIX may also intervene through the control of signaling pathways that modulate *Pax7* expression. Indeed, *Six1a* controls *Pax7* expression in myogenic stem cells through BMP signaling in zebrafish and the absence of *Six1a* leads to the exhaustion of PAX7+ cells [96]. The phenotype we observed is also reminiscent of that observed after the loss of Notch activity in the muscle lineage in which progenitor cells undergo massive differentiation and depletion of the progenitor pool [97–99]. Similarly, *Six2* plays a role in the amplification of nephron progenitors during embryogenesis [100,101] and its inactivation results in premature and ectopic differentiation of mesenchymal cells into epithelia and depletion of the progenitor cell population within the metanephric mesenchyme [37]. Finally, the requirement of *Six2* for the self-renewal of progenitor cells was also shown in the cartilage tissue whereby *Six2*-null newborn mice display premature fusion of the bones in the cranial base due to increased terminal differentiation of chondrocytes [78].

We and others have reported that SIX are important to activate *Myf5*, *MyoD*, *Myogenin* and many genes that are expressed in post-mitotic myonuclei of the myofibers [7]. While all the studies concerning the role of *Six* in the myogenic lineage during embryonic and adult myogenesis showed their requirement to efficiently activate *MyoD* and *Myog* [92,93], we show in this study that *Myog* can be activated independently of SIX in epaxial myogenic progenitors, and that the main upstream redundant role of SIX at this level is to allow their self-renewal and prevent their differentiation. The progressive acquisition by multipotent cells of more restricted fates, and the control of the balance between committed cell proliferation, and their differentiation associated with cell cycle arrest, are major aspects of myogenesis. SIX proteins are key regulators of these developmental steps, most probably through the recruitment of distinct transcription factors and cofactors to drive myogenic identity acquisition during embryonic development [7,66,73,93,94]. Our results are also in agreement with the role SIX1 and EYA1 proteins in synergy with ESRRB and PAX3 to activate *Pax7* expression in fibroblasts and reprogram these cells into an induced myogenic stem cell fate [102]. Our results, supported by a recent study demonstrating the involvement of SIX1 in the expansion of alveolar rhabdomyosarcoma cells and preventing their differentiation [103], show a dual role of SIX proteins in both, the acquisition and the maintenance of the myogenic stem fate, but also in their terminal differentiation through the control of many genes expressed in post mitotic differentiated cells. Finally, the characterization of the involvement of *Six1/2/4/5* genes in the

control of *Pax7* gene expression in muscle stem cells and in their self-renewal could allow the refinement of muscle stem cell activation pathways from hiPSCs [104,105].

## Methods

### Ethics statement

Animal experimentation was carried out in strict accordance with the European convention STE 123 and the French national charter on the Ethics of Animal Experimentation. Protocols were approved by the Ethical Committee of Animal Experiments of the Institut Cochin, CNRS UMR 8104, INSERM U1016, and by the Ministère de l'éducation nationale de l'enseignement et de la recherche, APAFIS#15699–2018021516569195.

### Mice and animal care

*Six5*$^{-/+}$ [38] and *Six2*$^{-/+}$ [37] mice were crossed with "*Six1*$^{-/+}$*Six4*$^{-/+}$" or with "*Six1*$^{-/+}$" mice to obtain "*Six1*$^{-/+}$*Six4*$^{-/+}$*Six2*$^{-/+}$*Six5*$^{-/-}$"; "*Six1*$^{-/+}$Six2$^{-/+}$"; "*Six2*$^{-/+}$*Six5*$^{-/-}$ or "*Six1*$^{-/+}$*Six4*$^{-/+}$Six2$^{-/+}$" adults. "*Six1*$^{-/-}$", "*Six1*$^{-/-}$*Six4*$^{-/-}$"; "*Six1*$^{-/-}$*Six4*$^{-/-}$*Six5*$^{-/-}$"; "*Six1*$^{-/-}$*Six4*$^{-/-}$*Six2*$^{-/-}$*Six5*$^{-/-}$"; "*Six1*$^{-/-}$*Six2*$^{-/-}$" and "*Six1*$^{-/-}$*Six4*$^{-/-}$*Six2*$^{-/-}$" mutants and their littermate control fetuses were obtained by crossing corresponding two- to four-month-old heterozygous mice respectively, backcrossed on C57BL/6N background: "*Six5*$^{-/-}$*Six1*$^{-/+}$*Six4*$^{-/+}$*Six2*$^{-/+}$" males and females were viable and fertile. "*Six1*$^{-/-}$*Six2*$^{-/-}$"; "*Six1*$^{-/-}$*Six4*$^{-/-}$*Six2*$^{-/-}$"; "*Six5*$^{-/-}$*Six1*$^{-/-}$*Six4*$^{-/-}$*Six2*$^{-/-}$"E18.5 fetuses were obtained at a Mendelian ratio of 1/16, excluding embryonic early lethality. *Pax3*$^{GFP/+}$ animals are from [20], and "*Six2*$^{-/-}$::*Six1*$^{flox/flox}$::*Wnt1-CRE*" have been described already [56]. Mice were maintained at temperature 22+/-2˚C, with 30 to 70% humidity and with a dark/light cycle of 12h/12h.

### Fetus preparation

Fetuses were staged, taking the appearance of the vaginal plug at embryonic day (E) 0.5. Harvested 10.5-, 14.5- or 18.5-days post-fertilization, fetuses were decapitated, and their skin was removed (except for qKO fetuses). They were fixed in 4% PFA for 15 min (E14.5) or 30 min (E18.5) at room temperature (RT) and kept in 15% sucrose-PBS at +4˚C overnight (ON) then embedded into OCT and snap frozen in isopentane (-30˚C), cooled in liquid nitrogen and kept at -80˚C until use. E10.5 embryos were fixed with 0.2% PFA ON at +4˚C then kept in 15% sucrose-PBS ON at +4˚C, embedded in OCT and frozen on dry ice the next day and kept at -80˚C until use. Transverse sections (10–12μm thickness; cryostat) were transferred to positively charged-slides (SuperFrost-plus; Thermo Fisher Scientific) and kept at -80˚C until use.

### Immunohistochemistry

Fetus sections were rehydrated in PBS before antigen retrieval treatment in a pH6 citrate buffer solution at 95˚C for 15 min followed by 20 min of cooling. They were blocked with 0.5% Triton complemented with 5% horse serum for 1-3h at RT. Sections were incubated with primary antibodies at +4˚C overnight, then with secondary antibodies for 45min-1h at RT; antibodies were diluted in the blocking solution. For *s1s4s2s5*KO fetuses, Pax7 immunostaining required different steps: sections were permeabilized with cold -20˚C acetone for 10 min, air-dried 10 min, then blocked and incubated with antibodies as previously explained. Six1 immunostaining required an amplification step using a biotinylated secondary antibody. Sections were incubated with horseradish peroxidase (HRP)-conjugated streptavidin for 30 min and treated with Alexa Fluor 488 tyramide for 10 min (SuperBoost tyramide signal amplification kit; Thermo Fisher Scientific). Immunostained sections were mounted in Mowiol mounting medium before imaging. Images were taken on either an upright fluorescent microscope

**Table 1. Primary antibodies used in this study.**

| Primary Antibody | Species | Company | Reference | Dilution |
|---|---|---|---|---|
| B galactosidase | Chicken | Abcam | ab134435 | 1/250 |
| Pitx2 | Rabbit | Abcam | ab221142 | 1/1000 |
| CD31 | Rat IgG2a | Santa Cruz | sc-18916 | 1/100 |
| a-SMA | IgG2a | Sigma Aldrich | 6228 | 1/500 |
| FoxC1 | Rabbit | Abcam | ab227977 | 1/100 |
| Pax3 | IgG2a | DSHB | | 1//20 |
| Ki67 | Rabbit IgG | Abcam | ab15580 | 1/100 |
| Laminin | Rabbit IgG | Sigma Aldrich | L9393 | 1/100 |
| Mf20 | Mouse IgG2b | DSHB | 3,601 | 1/100 |
| MYH3 | IgG1 | DSHB | F 1,652 | 1/100 |
| MYH7 | Mouse IgG1 | Sigma Aldrich | M8421 | 1/1000 |
| MYH8 | IgM | DSHB | N3-36 | 1/100 |
| Myogenin | IgG1 | Santa Cruz | sc-12732 | 1/100 |
| Pax7 | Mouse IgG1 | Santa Cruz | sc81648 | 1/100 |
| Pax7 | Ginea Pig IgG | | Brohl et al., 2012 | 1/1000 |
| Six1 | Rabbit | Sigma Aldrich | HPA001893 | 1/200 |
| Six2 | Rabbit | Abcam | ab68908 | 1/100 |
| Calponin | Rabbit | Abcam | ab46794 | 1/100 |
| FoxP2 | Mouse IgG1 | Santa Cruz | sc-517261 | 1/200 |
| MyoD | Rat IgG2a | Active Motif | 39991 | 1/100 |
| CyclinD1 | Rabbit | Abcam | ab134175 | 1//50 |
| Myf5 | Rabbit | Santa Cruz | sc-302 | 1/100 |
| Desmin | Rabbit | Abcam | ab32362 | 1/100 |

(Olympus BX63), equipped with an ORCA-Flash4.0 LT Hamamatsu camera, using Metamorph 7 software or with an inverted fluorescent confocal microscope (IXplore Spinning IX83) equipped with a Hamamatsu sCMOS Orca flash 4.0 V3 camera and using CellSens Dimension software. See Table 1 for primary antibodies references.

## Gel mobility shift assays

*In vitro* synthesis of SIX1, SIX2, SIX4 and SIX5 were obtained by the transcription/translation T7 TNT quick coupled transcription/translation kit (Promega). Gel mobility shift assays (GMSA) were performed as previously described [106], using *Myogenin* MEF3 double stranded DNA labelled probe. An excess of three hundred fold unlabelled DNA probes were added as competitor. See Table 2 for the sequences of DNA probes used.

## Mouse primary embryonic fibroblasts transfections

*SixqKO* (*Six1$^{-/-}$Six4$^{-/-}$::Six2$^{-/-}$::Six5$^{-/-}$*) mouse embryonic fibroblasts (MEFs) were isolated from the ventral skin of E13.5 embryos and cultured until naturally immortalized. Fibroblasts were cultured in Dulbecco's modified Eagle's medium (DMEM with 1g/l glucose; Invitrogen), supplemented with 10% fetal bovine serum (FBS; Invitrogen) and 0,1% penicillin–streptomycin (Invitrogen). C2C12 myoblasts were cultured in Dulbecco's modified Eagle's medium (DMEM with 1g/l glucose; Invitrogen), supplemented with 17% fetal bovine serum (FBS; Invitrogen) and 0,1% penicillin–streptomycin (Invitrogen). All cultures were grown in humidified incubators at 37˚C under 5% $CO_2$. Cells were transfected with lipofectamin 2000 (Invitrogen) with expression vectors for Six1-VP16, and reporter vectors BA-Myf5-Luc, tk-Luc,

**Table 2. Sequences of the oligonucleotides used for GMSA experiments.**

|  | Forward | Reverse |
|---|---|---|
| Myog MEF3 | 5'AGGGGGGC*TCAGGTTTC*TGTGGCGA | 5'CGCCACAGAAACCTGAGCCCCC |
| Pax3-59 | 5'ATGTCCC*TATTAATACA*GTATC | 5'GATACTGTATTAATAGGGA |
| Pax3-76 | 5'CATATGCAGA*TATCGATCCTTATCC*AAACTGATACGCTGA | 5'TCAGCGTATCAGTTTAAAGGATAAAGATCGATATCTG |
| Pax3-131 | 5'AGAA*TCAATTAGCCATGC*AGATTA | 5'AGGAAATGTAATCTGCATGGCTAATTGATTCT |
| Pax3-163 | 5'CCAATGTT*AATCTTTCAGACT*CAGAGCCGGTGAT | 5'ATCACCGGCTCTGAGTCTGAAAGATTAA |
| Pax3-211 | 5'CCAGATAGTA*TTAGATTT*CGTT | 5'AACGAAATCTAATACTAT |
| Pax7+140–1 | 5'AGAGCCAC*TCAGGTTT*CAGGATCCA | 5'TGGATCCTGAAACCTGAGTGGCT |
| Pax7+140–2 | 5'AGCCTGAGTAGGAACCTGAGGATACGG | 5'CCGTATCC*TCAGGTTCC*TACTCAGG |
| Myf5-S | 5'TGGCAGCGG*TCCAGTTTC*TCACAGA | 5'TCTGTGAGAAACTGGACCGCTG |
| Myf5-P | 5'TGGCTGTGAGAAACTGGACCGCTGC | 5'GCAGCGG*TCCAGTTTC*TCACAG |
| Pax3-253 | 5'AGAAGAAAGGCTCTCTGAAGCGTATTC | 5'GAATACGCT*TCAGAGAGC*C |
| MyogNF1 | 5'AGTATCTCTGGGTTCATGCCAGCAGGG | 5'CCCTGCTGGCATGAACCCAGAGATA |
| Rspo3 NKX | 5' AGATCCAGC*TCAAGTAGC*CTGGAT | 5'ATCCAGGCTACTTGAGCTGGAT |
| MEF3PAX7+60–1 | 5'AGGCTCTG*TCAAGCCTC*AGACCT | 5'AGGTCTGAGGCTTGACAGAGC |
| MEF3PAX7+60–2 | 5'AGCACCCTGCAAGCCAAGCCACA | 5'TGTGGC*TTGGCTTGC*AGGGTG |
| MEF3PAX7+60–3 | 5'AGAAGGGCGAAAGCAGAGACATA | 5'TATGTC*TCTGCTTTCGC*CCTT |
| MEF3PAX7+60–4 | 5'AGCAGGGC*TCTGTTTT*CACCACT | 5'AGTGGTGAAAACAGAGCCCTG |
| MEF3PAX7+60–5 | 5' AGCGTGGA*TCAGACCTC*CAGAGT | 5'ACTCTGGAGGTCTGATCCACG |
| MEF3PAX7+60–6 | 5'AGCCCCTGGAGAACAGAGCAATG | 5'CATTGC*TCTGTTCTC*CAGGGG |
| MEF3PAX7+60–7 | 5'AGAAAACTGACCCTTGACCCCAG | 5'CTGGGG*TCAAGGGT*CAGTTTT |
| MLC Hox | 5'AGACCTTA*TTAAATTAC*CATGTG | 5'CACATGGTAATTTAATAAGGT |
| HRE-EPO | 5' AGGCCCTACGTGCTGTCTCA | 5'TGAGACAGCACGTAGGGC |

**Table 3. Sequences of the oligonucleotides used to amplify the *Myf5* and *Pax7* enhancers.**

|  | Forward | Reverse |
|---|---|---|
| Myf5 BA | 5'TTTTGGTACCGTTTCAAAGACCTGTCTGC | 5'TTTTCTGCAGCAGTTTCTCACAGAAAGCTC |
| Pax7 +140 | 5'GGCCGTATCCTCAGGTTCCT | 5'GCACACAGTGGCTGGTGACT |
| Pax7 +40 | 5'CTCGAGCCATCAAAAGCTTGCCGACC | 5'CTCGAGGCGTCTGTGCTGTGACTACT |
| Pax7 +60 | 5'CTCGAGAAGGGCCAGCAGATTGTACC | 5'CTCGAGGCCCCCAGTTAAAGCCAGAT |

+40-Pax7-Luc or +60-Pax7-Luc in the presence of tk-Renilla internal control. See Table 3 for the sequences of the Luciferase reporters used. The medium was changed six hours after transfection, and forty-two hours after, cells were lysed. Luciferase and Renilla activities were measured with Centro XS3 LB 960 Luminoskan plate reader. The results shown are normalized by the tk-Renilla activity, then by the ratio between the specific activity and that observed for the Luc reporter with no enhancer.

## Supporting information

**S1 Fig. PAX3 and SIX expression at the limb buds levels. (A)** Immunostaining on transverse sections of E11.5 Pax3$^{GFP/+}$ embryos at the trunk level showing Pax3 (green), Six1 and Six2 (grey) and Dapi (blue); Upper panel "a" showing the mesenchymal cells and "b" showing the migrating progenitor cells to the limbs; FL: Forelimb, Sb = 200μm. Lower panel: zoom on the migratory progenitor cells "b", Sb = 50μm. **(B)** Immunostaining on E10.5 Ctrl (s1 Hz) and s1KO (n = 1) embryos transverse sections at the limb buds level for ß-gal (green), Pax3 (red) and Dapi (blue); upper panel zoom represents the dashed white square; NT: Neural Tube, HL:

Hindlimb; Sb = 50μm. **(C)** Immunostaining on E10.5 Ctrl (s1 Hz) and s1KO (n = 1) embryos transverse sections at the limb buds level for ß-gal (green), Pax3 (red), FoxC1 (cyan) and Dapi (blue). Lower panels represent a zoom of the Pax3 positive regions. HL: hindlimb, NT: neural tube; Sb = 50μm.
(TIFF)

**S2 Fig.** *Myf5-BA* **enhancer activity is enhanced by SIX1-VP16. (A-B)** Open chromatin regions shown by SIX1 and SIX4 ChIP-seq experiments at the Pax3 and Pitx2 regions, respectively. **(C)** Gel Mobility-Shift Assay (GMSA) using *Myogenin* MEF3 double stranded DNA probe with *in vitro* synthesized SIX1, SIX2, SIX4 and SIX5 and hundred-fold molar excess of indicated DNA competitor. See Table 2 for Pax3, Pax7, Myf5 and Myog DNA probes sequences. **(D)** Luciferase assays on mouse embryonic fibroblasts (MEF) qKO showing a significant activation of the poly-Mef3 promoter and of Myf5 branchial arch enhancer with Six1-VP16 chimeric protein; Statistical non-parametric t- test with mean ±s.e.m and **p<0.005, ****p<0.0001 **(E-F)** Open chromatin regions shown by SIX1 and SIX4 ChIP-seq experiments at the Myf5 and c-met regions respectively. Red rectangle for SIX4-ChIP-seq corresponds to Myf5 BA enhancer.
(TIFF)

**S3 Fig. Impaired hypaxial myogenesis in compound *Six* mutant embryos. (A)** Immunostaining on transverse sections of E18.5 WT and s2KO (n = 2) fetuses at the distal forelimb (left) Sb = 500μm and distal hindlimb (right) Sb = 500μm for sarcomeric myosins marked by MF20 (green) and Dapi (blue); U: Ulna, R: Radius, FDS: flexor digitorium sublimis, FDP: flexor digitorium profundus, Sup: supinator, F: Fibula, T: Tibia, TA: tibialis anterior, EDL: extensor digitorum longus, S: Soleus, Gas: Gastrocnemius. **(B)** Immunostaining on transverse sections of E18.5 WT, s1s4dKO (n = 3) and s1s2dKO (n = 2) fetuses at the trunk level for sarcomeric myosins marked by MF20 (green) and Dapi (blue); FL: Forelimb, H: Heart, NT: Neural Tube, Sb = 500μm. **(C)** Immunostainings on E18.5 Ctrl and s1s2dKO (n = 2) fetuses transverse sections on the distal hindlimbs level for sarcomeric myosins marked by MF20 (green) and Dapi (blue); T: Tibia, F: Fibula, Sb = 200μm.
(TIFF)

**S4 Fig. PAX7+ cells in muscles of compound *Six* fetuses. (A)** Immunostainings on E18.5 WT fetuses on transverse sections at the head level showing the EOM marked by MF20 (green), and the Pax7 (red) cells expressing Six1 or Six2 or Pitx2 (grey) and Dapi (blue); E: eye, Sb = 150μm (upper panel) Sb = 20μm (lower panel). **(B)** Immunostainings on E18.5 WT, s2s5dKO (n = 2) and s1s4s5tKO (n = 2) fetuses at the head level showing the EOM (upper panel) and the masseter muscles (lower panel), Pax7 (green), MF20 (red) and Dapi (blue); Sb = 15μm. **(C)** Immunostainings on E18.5 WT and s1s2s4tKO fetuses on frontal sections at the head level showing the EOM marked by MF20 (green), and Dapi (blue); E: eye, Sb = 500μm.
(TIFF)

**S5 Fig. Calponin and Myosin heavy chain expression in E10.5 s1s2 dKO embryos at the craniofacial level. (A)** Immunostainings on E10.5 Ctrl (s1 Hz), s1s2dKO (n = 3) embryos transverse sections at the branchial arches (BA) level for ß-gal (purple), Calponin (green), MF20 (red) and Dapi (blue); Sb = 200μm. **(B)** Immunostainings on E10.5 Ctrl (s1 Hz), s1s2dKO (n = 3) embryos transverse sections at the head (EOM) level for ß-gal (purple), Calponin (green), MF20 (red) and Dapi (blue); E: Eye, Sb = 200μm.
(TIFF)

**S6 Fig. Esophagus muscles in compound *Six1/Six2* mutant fetuses.** Immunostainings on E18.5 Ctrl (s1 Hz), s1KO::s2Hz (n = 2), and s1Hz::s2KO (n = 2) fetuses at the trunk level showing the esophagus stained with MF20 (green), Desmin (red) and ß-gal (grey); oe: esophagus, Sb = 40µm.
(TIFF)

**S7 Fig. SIX2 is detected in PAX7+ cells. (A)** Upper panel: immunostainings on transverse sections of WT and s1s2s4tKO E14.5 fetuses showing on the left the whole fetus with sarcomeric myosins marked by MF20 (green) and Dapi (blue); FL: Forelimb, H: heart, NT: Neural Tube, Sb = 300µm, and on the right a zoom on the dorsal muscle masses stained with Pax7 (green), MF20 (red) and Dapi (blue); Sb = 20µm. Lower Panel: immunostainings on transverse sections of WT and s1s4s5tKO E18.5 fetuses showing on the left the whole fetus with sarcomeric myosins marked by MF20 (green) and Dapi (blue); Sb = 300µm, and on the right a zoom on the dorsal muscle masses stained with Pax7 (green), MF20 (red) and Dapi (blue); Sb = 15µm. **(B)** Immunostainings on E14.5 WT fetuses transverse sections at the trunk level showing on dorsal muscle masses Pax7 (green) and Six2 (red); white arrowheads indicate Pax7 +Six2+ cells and yellow arrowheads indicate Pax7+Six2- cells, Sb = 20µm. **(C)** Left: immunostainings of E14.5 Ctrl and qKO (n = 3) fetuses at the trunk level showing dorsal muscles masses staining for Pax7 (green), CyclinD1 (grey) and Dapi (blue); Sb = 10µm. Right: Quantification of the percentage of Pax7+ CyclinD1+ cells in the dorsal muscle masses of E14.5 Ctrl and qKO (n = 3) fetuses. Welch's statistical test with mean ±s.e.m; ns: non-significant with p = 0.13.
(TIFF)

**S8 Fig. Remaining neck muscles in E14.5 *Six* qKO.** Immunostaining on a cross-section of E14.5 Ctrl and qKO fetuses at the trunk level showing MF20 (red), Islet1/2 and FoxP2 respectively (green) and Dapi (blue) Sb = 500µm. Zoom on the lower panels represent the dashed white squares and Sb = 200µm.
(TIFF)

**S9 Fig. Opened DNA regions at the *Pax7* locus. (A)** snATAC-seq experiments with adult skeletal muscles at the Pax7 locus showing opening chromatin at +140, +60 and +40Kb regions. **(B)** Six1-ChIP-seq and Six4-ChIP-seq experiments at the Pax7 locus. **(C)** Gel Mobility-Shift Assay (GMSA) using *Myogenin* MEF3 double stranded DNA probe with *in vitro* synthesized SIX1, SIX2, SIX4 and SIX5 and hundred-fold molar excess of indicated DNA competitor. See Table 2 for Pax7, NF1, RSPO3 and HOX DNA probes sequences. **(D)** Luciferase assays in mouse C2C12 cells showing a significant activation of the Pax7+40 and Pax7+60 enhancers with Six1-VP16 chimeric protein. Statistical non-parametric t- test with mean ±s.e. m and **p<0.005 and *p<0.05.
(TIFF)

**S10 Fig. Table recapitulating the myogenic phenotype of different *Six* mutant and compound mutant embryos, as shown by this study and others.**
(JPG)

# Acknowledgments

We thank S.Dietrich for helpful critical reading of the manuscript. We thank A.Sotiropoulos for helpful discussions and J.Demignon who initiated this study. We thank P.Bourdoncle and T.Guilbert of the IMAG'IC facility, F.Letourneur and B.Saintpierre of the GENOM'IC facility and R.Pierre and M. Do-Cruzeiro of the MOUST'IC core facility and A.Guéraud of the

Institute Cochin for their technical assistance. We thank O. Brégerie, M. Mihoc and J. Perron-net at the UMS 28-105B of UPMC.

## Author Contributions

**Conceptualization:** Maud Wurmser, Rouba Madani, Stéphanie Backer, Ramkumar Sambasivan, Pascal Maire.

**Formal analysis:** Maud Wurmser, Rouba Madani, Nathalie Chaverot, Stéphanie Backer, Matthieu Dos Santos, Marc Santolini, Pascal Maire.

**Investigation:** Maud Wurmser, Rouba Madani, Nathalie Chaverot, Stéphanie Backer, Matthew Borok, Glenda Comai, Ramkumar Sambasivan, Rulang Jiang, Pascal Maire.

**Methodology:** Maud Wurmser, Rouba Madani, Stéphanie Backer, Matthieu Dos Santos, Ramkumar Sambasivan, Pascal Maire.

**Supervision:** Pascal Maire.

**Writing – original draft:** Maud Wurmser, Rouba Madani, Pascal Maire.

**Writing – review & editing:** Maud Wurmser, Rouba Madani, Glenda Comai, Shahragim Tajbakhsh, Frédéric Relaix, Pascal Maire.

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
