## [Decision Letter · Decision Letter 0]

7 Nov 2022

Dear Dr Maire,

Thank you very much for submitting your Research Article entitled 'Overlapping functions of SIX homeoproteins during embryonic myogenesis' to PLOS Genetics.

The manuscript was fully evaluated at the editorial level and by independent peer reviewers. The reviewers appreciated the attention to an important problem, but raised some substantial concerns about the current manuscript. Based on the reviews, we will not be able to accept this version of the manuscript, but we would be willing to review a much-revised version. We cannot, of course, promise publication at that time.

If you decide to revise the manuscript for further consideration at PLOS Genetics, please aim to resubmit within the next 60 days, unless it will take extra time to address the concerns of the reviewers, in which case we would appreciate an expected resubmission date by email to plosgenetics@plos.org.

We are sorry that we cannot be more positive about your manuscript at this stage. Please do not hesitate to contact us if you have any concerns or questions.

Yours sincerely,

Gregory A. Cox

Academic Editor

PLOS Genetics

Gregory P. Copenhaver

Editor-in-Chief

PLOS Genetics

Reviewer's Responses to Questions

**Comments to the Authors:**

Reviewer #1: The authors employ complex double, triple and quadruple KO in mice, of combinations of SIX proteins to examine their roles in myogenesis. The study examines craniofacial and epaxial neck muscles. The roles of SIX proteins have been investigated extensively and in this reviewers opinion the advances are somewhat incremental, the broader significance of the study is less clear. There is missing information which could contribute additional insights.

For example, SIX are involved in craniofacial myogenesis, and Myf5 expression fails to be activated. The alternative fate of the cells should be characterised further (page 7, Fig. 2B). The authors suggest they may become smooth muscle cells, this could be assessed using relevant markers.

The also find that quadruple KO fetuses do form neck muscles and myogenesis is activated in this context. There is no functional explanation for this important difference.

There are fewer fibres and epaxial muscle mass does not increase. A possible mechanistic explanation suggested are effects on Pax7 enhancer elements identified previously by the lab using snATAC seq. In vitro GMSA and transient transfection of reporters in C2C12 cells support a possible interaction with a constitutively active SIX-VP16 protein. Effects on the expression of the Pax7-GFP transgene (BAC) could have been examined in vivo in the context of various SIX KOs to confirm the cell-based experiments.

Based on immunostaining and cell counts the authors suggest that PAX7 positive cells differentiate and thus the progenitor population becomes depleted. This is very difficult to see in the panels and not convincing,

PAX7+MYOG+Ki67- cells in the qKO need to be more clearly identified.

Reviewer #2: Using a variety of single and compound mutants, Wurmser et al refine the genetic network of SIX genes in the control of craniofacial as well as epaxial and hypaxial trunk myogenesis.

The authors show a combinatorial requirement of Six1 and Six4 for a) proper Pax3 expression via direct regulation, and importantly show that in absence of both, Pax3+ cells are deviated from the myogenic to an endothelial and smooth muscle fate, thereby also leaving their original migration trajectory. In craniofacial muscles, Six1 and Six2 were strictly required for myogenesis, and importantly the authors show that in this context Pitx2 expression was still present, suggesting that Pitx2 alone was unable to drive myogenesis. Finally, the authors show a strict requirement for all four SIX proteins in fetal epaxial myogenesis and perinatal MYH expression, as well as the maintenance of a PAX7+ pool, likely by direct control of Pax7 expression.

The work is very well executed and the authors used an impressive array of compound mutants to dissect the requirement of SIX genes in individual muscle groups of distinct embryonic origin. Although most of the conclusions are backed by the data presented, in some instances a better data visualization or quantification of immunolabeling data would strengthen the data. There are a few point the authors should address:

1. In Fig. S1B it is very hard to appreciate normal muscle patterns in Six2 KO fetuses in the images provided. Sections of comparable levels and orientation should be shown. Also, it would be nice to label the muscles.

2. Since Six2 is also expressed locally in limb mesenchyme, smaller phenotypes caused by local mesenchyme-myoblast interactions (as e.g. in the Tbx3 KO with only few muscles affected, DOI: 10.1242/dmm.025874) may have been overlooked. Ideally, ideally several sections at different levels should be shown to detect such changes. However, I appreciate that this is not the focus of the manuscript

3. The authors claim (p6) that “we observed … few Six1-β-Gal+ s1s2 mutant cells at the dorsal aorta level, expressing α-SMA or CD31”, which likely is supposed to be compared to many of such cells in s1/s4 DKOs; however, on the images in Fig. 1A, I do not see differences between s1/s4 or s1/s2 embryos. A quantification should be provided.

Along the same line, the authors find “few PAX3+ cells in limbs of s1/s2 embryos; are these indeed fewer than in controls, this should be quantified. Are the sections shown in Fig. 1B on fore- or hindlimb levels? Both should be shown, as muscle formation in forelimbs appears abrogated, while being affected less severely in hindlimbs (although this is not properly displayed; sections as in Fig. S1C should be shown for the hindlimb level as well to appreciate proximal muscles).

4. In s1/s4 DKOs the authors have demonstrated previously lack of c-met expression; for the sake of completeness, the authors may want to show this for the s1/s2 mutants as well.

5. Fig. S3B and Fig. 5B should be quantified

6. In Fig. 3B, the authors use Desmin expression as an indicator of a possible switch to a smooth muscle fate of EOM muscle progenitors; it would be more appropriate to use alpha-SMA immunostaining as the authors have used in the trunk.

Reviewer #3: In this paper, the authors studied the roles of different SIX homeoproteins in the control of myogenic cell behaviours in different anatomical locations during mouse embryogenesis. Precisely, they investigate the overlapping roles of Six1, Six2, Six4 and Six5 in the craniofacial muscles and HL muscles.

The paper is overall interesting as it further documents the role of SIX homeoproteins in myogenesis.

My main comment is that the authors are sometimes making quantitative assumptions based on immunostainings panels without quantification to sustain their conclusions.

Precisely; in Figure 1C, the authors claim that “An increased FOXC1 expression was also detected in mutant b-Gal+ hypaxial cells of the dermomyotome of E10.5 s1s2dKO at the hindlimb level, where a few of those b-Gal+ s1s2dKO cells expressed FOXC1”.

Such statement could be supported by a quantification of the number of FOXC1 positive cells in the dermomyotome, as well as a quantification of the number of FOXC1+/�-Gal positive cells in Ctrl, s1s4 KO and s1s2 KO samples.

In Figure S3B, the authors claim that “While the masseters, derived from the first branchial arch, formed normally in s1s4dKO, they were hypoplasic in s1s4s5tKO. Furthermore, both s1s4s5tKO and s2s5 dKO muscles masses show the presence of less PAX7+ cells (Fig.S3B).

There is no data to support the hypoplasia of the Masseter in s1s4s5tKO, and the assumption that the number of Pax7+ cells is reduced is not supported by quantitative data. Furthermore, the panels of immunostaining for Pax7 in the Masseter of WT, s2s5 KO and s1s4s5tKO is not convincing enough to make the above statement.

In Figure 5B, the authors mention that “As suspected, qKO fetuses showed an increase in embryonic MYH (MYH3) as well as slow MYH (MYH7) positive myofibers, while no fast perinatal MYH (MYH8) was detected in their remaining epaxial muscle masses (Fig. 5B).

A quantification of the number of positive fibres could be carried out to support this statement.

In Figure S5C, the authors declare “Accordingly, we observed a decreased percentage of PAX7+ CyclinD1+ cells in the qKO E14.5 fetuses in epaxial muscles (Fig.S5C).”

The graph shows a trend, however, the quantification of the number of Pax7+ CyclinD1+ cells in the qKO versus control does not indicate statistical significance. Could the author provide the p value? Could they strengthen this data with a staining for Ki67, as they did in the final figure?

The role of Six5 remains elusive and could be more broadly discussed. Have the authors investigated the expression of Six5 in the presomitic mesoderm? The authors have shown in this study and previously that a proportion of Six1 and Six2 cells are co-expressing Pax3 in the dermomyotome, what about the expression of Six5 in this compartment?

To help recapitulate their findings, the authors could generate a schematic that illustrate the different roles of Six genes in myogenesis, confirmed and presumed by this study.

**Have all data underlying the figures and results presented in the manuscript been provided?**

Reviewer #1: None

Reviewer #2: **No: **I could not find numerical data underlying Figs 5A, C; 6A, B; S2D; S5C; S6D

Reviewer #3: Yes

PLOS authors have the option to publish the peer review history of their article (what does this mean?). If published, this will include your full peer review and any attached files.

Reviewer #1: No

Reviewer #2: No

Reviewer #3: No

---

## [Decision Letter · Decision Letter 1]

10 May 2023

Dear Dr Maire,

We are pleased to inform you that your manuscript entitled "Overlapping functions of SIX homeoproteins during embryonic myogenesis" has been editorially accepted for publication in PLOS Genetics. Congratulations!

Yours sincerely,

Gregory A. Cox

Academic Editor

PLOS Genetics

Gregory P. Copenhaver

Editor-in-Chief

PLOS Genetics

Comments from the reviewers (if applicable):

Reviewer's Responses to Questions

**Comments to the Authors:**

Reviewer #1: The modifications introduced in the text did not appear in red as indicated in the response. This would have been helpful.

The authors have improved labelling of figures, added some quantification and a summary figure, which has improved the manuscript.

Reviewer #2: The authors have addressed all my concerns, congratulations to this very nice piece of work!

Reviewer #3: We acknowledge the substantial revision of the manuscript to our and all the reviewers comments. We have no further comments and our questions have been expertly answered and attended to. We look forward to seeing the manuscript in print.

**Have all data underlying the figures and results presented in the manuscript been provided?**

Reviewer #1: Yes

Reviewer #2: Yes

Reviewer #3: Yes

PLOS authors have the option to publish the peer review history of their article (what does this mean?). If published, this will include your full peer review and any attached files.

Reviewer #1: No

Reviewer #2: No

Reviewer #3: No

**Data Deposition**

http://datadryad.org/submit?journalID=pgenetics&manu=PGENETICS-D-22-01142R1

**Press Queries**

---

## [Editor Report · Acceptance letter]

30 May 2023

PGENETICS-D-22-01142R1 

Overlapping functions of SIX homeoproteins during embryonic myogenesis 

Dear Dr Maire, 

We are pleased to inform you that your manuscript entitled "Overlapping functions of SIX homeoproteins during embryonic myogenesis" has been formally accepted for publication in PLOS Genetics! Your manuscript is now with our production department and you will be notified of the publication date in due course.

With kind regards,

Zsofi Zombor

PLOS Genetics

On behalf of:
